# PQBP3 prevents senescence by suppressing PSME3-mediated proteasomal Lamin B1 degradation

Yuki Yoshioka [ID][1,5], Yong Huang[1,5], Xiaocen Jin[1,5], Kien Xuan Ngo [ID][2,5], Tomohiro Kumaki[1], Meihua Jin[1], Saori Toyoda[1,3], Sumire Takayama[1], Maiko Inotsume[1], Kyota Fujita[1,4], Hidenori Homma[1], Toshio Ando [ID][2], Hikari Tanaka[1,5] & Hitoshi Okazawa [ID][1✉]

## Abstract

Senescence of nondividing neurons remains an immature concept, with especially the regulatory molecular mechanisms of senescence-like phenotypes and the role of proteins associated with neurodegenerative diseases in triggering neuronal senescence remaining poorly explored. In this study, we reveal that the nucleolar polyglutamine binding protein 3 (PQBP3; also termed NOL7), which has been linked to polyQ neurodegenerative diseases, regulates senescence as a gatekeeper of cytoplasmic DNA leakage. PQBP3 directly binds PSME3 (proteasome activator complex subunit 3), a subunit of the 11S proteasome regulator complex, decreasing PSME3 interaction with Lamin B1 and thereby preventing Lamin B1 degradation and senescence. Depletion of endogenous PQBP3 causes nuclear membrane instability and release of genomic DNA from the nucleus to the cytosol. Among multiple tested polyQ proteins, ataxin-1 (ATXN1) partially sequesters PQBP3 to inclusion bodies, reducing nucleolar PQBP3 levels. Consistently, knock-in mice expressing mutant *Atxn1* exhibit decreased nuclear PQBP3 and a senescence phenotype in Purkinje cells of the cerebellum. Collectively, these results suggest homologous roles of the nucleolar protein PQBP3 in cellular senescence and neurodegeneration.

**Keywords** PQBP3; Lamin B1; Senescence; Neurodegeneration; Nuclear Membrane Instability
**Subject Categories** Cell Cycle; Membranes & Trafficking; Neuroscience

## Introduction

The nucleolus has an indispensable role in cell viability and ribosomal RNA (rRNA) transcription from ribosomal DNA (rDNA) (Boisvert et al, 2007). Unlike other organelles, the nucleolus is not encapsulated by a lipid bilayer membrane but instead is an assembly of multiple nucleolar intrinsically disordered proteins (IDPs) together with RNAs, and is sequestered by liquid-liquid phase separation (LLPS) (Feric et al, 2016; Shin and Brangwynne, 2017; Mitrea et al, 2018; Jin et al, 2023). Based on electron microscopy findings, the nucleolus is further divided into the granular component (GC), fibrillar center (FC), and dense fibrillar component (DFC) substructures (O'Sullivan et al, 2013). Multiple proteins, such as nucleolin, fibrillarin, nucleophosmin (NPM1), and Pescadillo ribosomal biogenesis factor 1 (PES1), are localized to distinct substructures of the nucleolus. For example, fibrillarin is a marker of the DFC (Yao et al, 2019) while nucleolin, NPM, and PES1 are distributed in the GC (Biggiogera et al, 1990; Boulon et al, 2010).

Previously we aimed for identifying common disease modifier proteins in a group of nine neurodegenerative diseases called polyQ disease that share expansion of polyQ repeat in causative proteins due to CAG repeat expansion in their gene mutations (La Spada et al, 1994; Zoghbi and Orr, 2000), and discovered polyglutamine binding protein 3 (PQBP3) in yeast two-hybrid (Y2H) screening of the human brain cDNA library (Imafuku et al, 1998; Waragai, 1999). PQBP3 was later identified via proteomic analysis as a component of nucleolus and named nucleolar protein 7 (NOL7) (Ahmad et al, 2009). The other new polyQ-binding proteins discovered by Y2H screening include PQBP1 and PQBP5/NOL10. PQBP1 is an intrinsically disordered protein (IDP) (Rees et al, 2012; Takahashi et al, 2009; Mizuguchi et al, 2014; Okazawa, 2018) and, in the nucleus, regulates transcription and splicing of mRNAs encoding cell cycle regulators in neural stem cells and mRNAs encoding regulators of synapse function in mature brain neurons (Okazawa et al, 2002; Ito et al, 2015a, 2015b). Consistently, congenital mutations of the human *PQBP1* gene cause intellectual disability and microcephaly (Kalscheuer et al, 2003; Stevenson et al, 2005), and acquired PQBP1 dysfunction causes neuronal synapse loss in Alzheimer's disease (Tanaka et al, 2018). In the cytoplasm, PQBP1 functions as an intracellular receptor for human immunodeficiency virus 1 (HIV1) (Yoh et al, 2015, 2022) and tau (Jin et al, 2021), activating the cGAS-STING pathway and proinflammatory responses of innate immune cells (Yoh et al, 2015, 2022; Jin et al, 2021).

[1]Department of Neuropathology, Medical Research Institute, Tokyo Medical and Dental University, 1-5-45, Yushima, Bunkyo-ku, Tokyo 113-8510, Japan. [2]Nano Life Science Institute, Kanazawa University, Kakuma-machi, Kanazawa, Ishikawa 920-1192, Japan. [3]Department of Psychiatry and Behavioral Sciences, Tokyo Medical and Dental University Graduate School, 1-5-45, Yushima, Bunkyo-ku, Tokyo 113-8510, Japan. [4]Research Center for Child Mental Development, Kanazawa University, Kakuma-machi, Kanazawa, Ishikawa 920-1192, Japan. [5]These authors contributed equally: Yuki Yoshioka, Yong Huang, Xiaocen Jin, Kien Xuan Ngo, Hikari Tanaka. ✉E-mail: okazawa.npat@mri.tmd.ac.jp

PQBP5 was identified as a nucleolar protein by mass spectrometry analysis simultaneously with PQBP3/NOL7, and named nucleolar protein 10 (NOL10) (Ahmad et al, 2009). In our recent study, we delineated the physiological and pathological functions of PQBP5/NOL10 (Jin et al, 2023). Briefly, like PQBP1, PQBP5/NOL10 is also an IDP, and functions as an anchor protein to maintain the structure of the nucleolus; in the absence of PQBP5/NOL10, other nucleolar proteins are dispersed into the nucleoplasm (Jin et al, 2023). Moreover, PQBP5/NOL10 is sequestered to the inclusion bodies of polyQ disease proteins, which is the common mechanism of the polyQ category of neurodegenerative diseases (Perutz, 1999; Wanker, 2000; Chai et al, 2002; Stenoien et al, 2002; Kim et al, 2002). The resultant depletion of nucleolar PQBP5/NOL10 causes disappearance of the total nucleolus structure in brain neurons (Jin et al, 2023).

Genetic mutations of PQBP3/NOL7 have been implicated in various types of cancers, including melanoma, breast carcinoma, leukemia, lymphoma, osteosarcoma, retinoblastoma, nasopharyngeal carcinoma, and cervical cancer (Pinho et al, 2019; Hasina et al, 2006; Doçi et al, 2012; Li et al, 2021). Abundant evidence demonstrating a significant function of the nucleolus in the physiological and pathological roles of PQBP3/NOL7, as well as the involvement of PQBP1 and PQBP5/NOL10, prompted us to investigate other aspects of PQBP3/NOL7 function in the context of neuronal disease.

Senescence, characterized by disabled cell division in normal cells, discovered by Leonard Hayflick (Hayflick, 1965; Hayflick and Moorhead, 1961), is an aging-related phenotype (Di Micco et al, 2021; Gorgoulis et al, 2019; Yang and Sen, 2022). Deficiency of the generally accepted senescence marker Lamin B1 (Freund et al, 2012; Shimi et al, 2011; Wang et al, 2017) induces enlargement of the Lamin A/C mesh at the surface of the nuclear membrane and leakage of DNA from the nucleus (Shimi et al, 2011, 2008). Cytoplasmic genomic DNA, which is thought to originate from micronuclei generated by chromosomal segregation errors of unrepaired DNA damage or from DNA damage (Harding et al, 2017; Mackenzie et al, 2017) such as deficiency of ataxia-telangiectasia mutated (ATM) (Song et al, 2019), is considered to be a marker of senescence (Di Micco, 2017; Miller et al, 2021). The link between the nucleolus and senescence remains largely unknown, although early studies in yeast genetics suggested a role for the nucleolus in the senescence process (Guarente, 1997; Johnson et al, 1998). However, the relationship between the nucleolus and senescence has recently become a topic of intense investigation. In yeast cells, ribosomal DNA forms extrachromosomal rDNA circles (ERCs) under senescence, while the counterpart mechanisms in mammalian cells are now under investigation (Yang et al, 2015; Kasselimi et al, 2022). Moreover, intracellular Aβ accumulation in Alzheimer's disease (AD) or intracellular TDP43 accumulation in frontotemporal lobar degeneration (FTLD) is associated with senescence phenotypes (Tanaka et al, 2020; Homma et al, 2021), while the mechanism underlying the link between intracellular disease protein accumulation and senescence is not completely understood.

In the present study, we reveal that PQBP3/NOL7 is an IDP protein localized primarily to the periphery of the nucleolus. PQBP3/NOL7 localization and chromatin distribution were closely related, such that distribution of nuclear chromatin was altered and ultimately disappeared from the nucleus in the absence of nucleolar PQBP3/NOL7. Interestingly, exogenous overexpression of PQBP3/NOL7 suppressed cytoplasmic genomic DNA leakage in senescent cells (Di Micco, 2017; Miller et al, 2021; Lan et al, 2019).

Bioinformatics analyses using the protein-protein interaction (PPI) database String (version 11.5) suggested a direct interaction between PQBP3/NOL7 and the 11S proteasome activator complex subunit 3 (PSME3), which activates a trypsin-like catalytic subunit of the proteasome complex and facilitates degradation of p53/TP53 (Zhang and Zhang, 2008) or KLF2, a negative regulator of NF-κB transcriptional activity (Sun et al, 2016). Further, loss of the interaction between PQBP3 and PSME3 desuppressed degradation of Lamin B1. In the context of polyQ disease pathology, we revealed that nuclear PQBP3/NOL7 deficiency in Purkinje cells of the SCA1 mouse model ($Sca1^{154Q/2Q}$) in which an expanded repeat of 154 CAGs was targeted into the mouse Ataxin-1 (Atxn1) locus (Watase et al, 2002) is linked to their senescent phenotype.

# Results

## Distribution of PQBP3/NOL7 in the nucleolar periphery

First, we performed immunocytochemistry in HeLa cells to identify the detailed nucleolar localization of PQBP3/NOL7 (Fig. 1). Previous studies reported nucleolar localization of alternative PQBP3/NOL7 splicing products, concluding that full-length PQBP3/NOL7 is in the GC (Kinor and Shav-Tal, 2011). Another experiment overexpressing NOL7-GFP demonstrated signal halation in the nucleolus (Zhou et al, 2010). Using an anti-PQBP3/NOL7 polyclonal antibody generated against the full-length protein, we revealed that PQBP3/NOL7 was distributed in the periphery of the nucleolus, while fibrillarin was located in the central region of the nucleolus (Fig. 1A).

Super-resolution microscopy (SRM) further characterized the distribution pattern of PQBP3/NOL7 at the outer nucleolar shell (Fig. 1B). Nucleolin but not fibrillarin had a similar distribution pattern but at a higher density (Fig. 1B). In addition, small PQBP3/NOL7 speckles (250–300-nm diameter) were scattered homogenously throughout the nucleoplasm and cytoplasm (Fig. EV1a). These small speckles were detectable at the confocal microscopy level (Fig. 1A) but were confirmed definitively by SRM (Fig. 1B). A previous study reported that PQBP3/NOL7 splicing isoforms lacking the amino acids corresponding to the region from exon 5 to exon 8 are distributed in the nucleoplasm (Kinor and Shav-Tal, 2011). However, distribution of the shorter isoform was homogenous and did not exhibit the small speckle-like localization identified with an antibody against the full-length isoform in the present study. They did not detect small speckle of full-length PQBP3/NOL7 in the nucleoplasm and cytoplasm (Kinor and Shav-Tal, 2011), presumably due to the use of lower-resolution microscopy and the overexpression of a GFP fusion protein.

In addition, human iPSC-derived neurons were subjected to SRM to characterize the distribution pattern of PQBP3/NOL7 in terminally differentiated cells (Fig. 1C). PQBP3/NOL7 was located at the outer nucleolar shell in human iPSC-derived neurons similarly to HeLa cells (Fig. 1C).

## Negative correlation of quiescence with nucleolar localization of PQBP3/NOL7

In immunocytochemistry studies, we observed that nucleolar PQBP3/NOL7 signal decreased when HeLa cells become confluent and

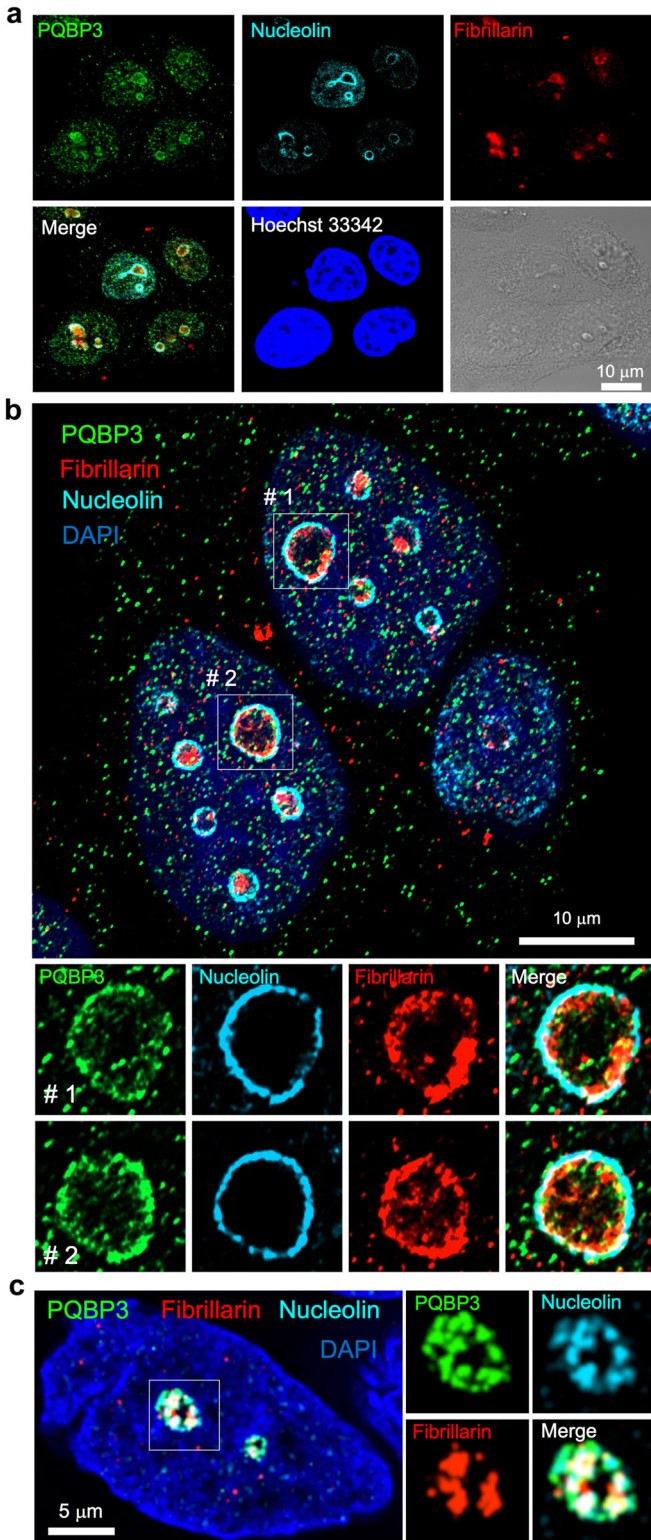

**Figure 1. PQBP3/NOL7 is a nucleolar protein located predominantly in the outer nucleolar shell.**

(A) Confocal microscopic analysis of unfixed HeLa cells penetrated by Tween20 and immunostained with anti-PQBP1, antifibrillarin, and antinucleolin antibodies. Nuclei were costained with Hoechst 33342. The signals of PQBP3/NOL7 were distributed in the peripheries of nucleoli. In addition, smaller speckles were observed in the nucleoplasm and cytoplasm. (B) Super-resolution microscopy images of HeLa cells after fixation. The distribution pattern was similar to that observed by standard confocal microscopy. The speckle diameters were ~260 nm in the nucleoplasm and ~300 nm in the cytoplasm. Staining localized to the outer shell of the nucleolus resembled a chain or cluster of similarly sized speckles. (C) Super-resolution microscopy images of normal iPSC-derived neurons after fixation. The relationship of PQBP3/NOL7, nucleolin, and fibrillarin was similar to that in HeLa cells. Source data are available online for this figure.

PQBP3/NOL7, 0: no nucleolar PQBP3/NOL7). These analyses detected a negative relationship between cell-to-cell contact and nucleolar PQBP3/NOL7 localization (Fig. 2B). In addition, quantitative analyses of signal intensities in immunohistochemistry and western blot analyses revealed that the signal intensities of both nuclear and total cellular PQBP3/NOL7 were decreased while cytoplasmic PQBP3/NOL7 was increased when cultured cells reached confluence (Figs. 2C and EV1b,c).

In addition, we observed a proportion of nondividing HeLa cells that decreased nucleolar PQBP3/NOL7, maintained nucleoli stained with fibrillarin, but revealed cytoplasmic DNA implicated in DNA damage (Harding et al, 2017; Mackenzie et al, 2017) and senescence (Song et al, 2019; Di Micco, 2017) (Fig. 2D). The morphology in electron microscopy (EM) revealed nuclear membrane fragile but not micronuclei existence of such abnormal cells (Fig. 2E). These findings collectively indicated the negative correlation between contact inhibition and nucleolar localization of PQBP3/NOL7 and suggested accumulation of DNA damage during the process, which further promoted us to further investigate the relationship between senescence and nucleolar PQBP3/NOL7.

## Negative correlation of senescence with nucleolar localization of PQBP3/NOL7

Senescence can be induced in HeLa cells by various manipulations (Goodwin and DiMaio, 2001). We experienced that HeLa cells slow down proliferation after ten passages (>10G, 3–4 cell divisions between passages) in our laboratory condition. In immunohistochemistry characterization of PQBP3/NOL7 localization, we observed that nucleolar PQBP3/NOL7 staining diminished after a large number of passages after twenty passages (Fig. EV2a, red arrows). In addition, after twenty passages in most of the cells lacking nucleolar PQBP3/NOL7, the chromatin was disproportionally distributed or almost disappeared in the nucleus (Fig. EV2b,c, red or purple arrows). During cell division, PQBP3/NOL7 exhibited a specific pattern of cytoplasmic distribution, localized predominately to the centrosome (Fig. EV2d–f, white arrow), which was distinct from the abovementioned cells. Quantitative analyses confirmed the increase of red arrow type and purple arrow type of cells during passages (Fig. EV2g). These data suggested relevance between PQBP3/NOL7 and senescence.

Therefore, we quantitatively analyzed the relevance. First, staining for beta-galactosidase (β-Gal), a representative marker of

proliferation was arrested (Fig. 2A). We utilized quantitative analysis of contact inhibition relative to the presence of nucleolar PQBP3/NOL7 (Fig. 2B). We quantified the percentage of cell margins in contact with neighboring cells and compared this percentage with the presence of nucleolar PQBP3/NOL7, as estimated qualitatively by three stages (1: robust nucleolar PQBP3/NOL7, 0.5: faint nucleolar

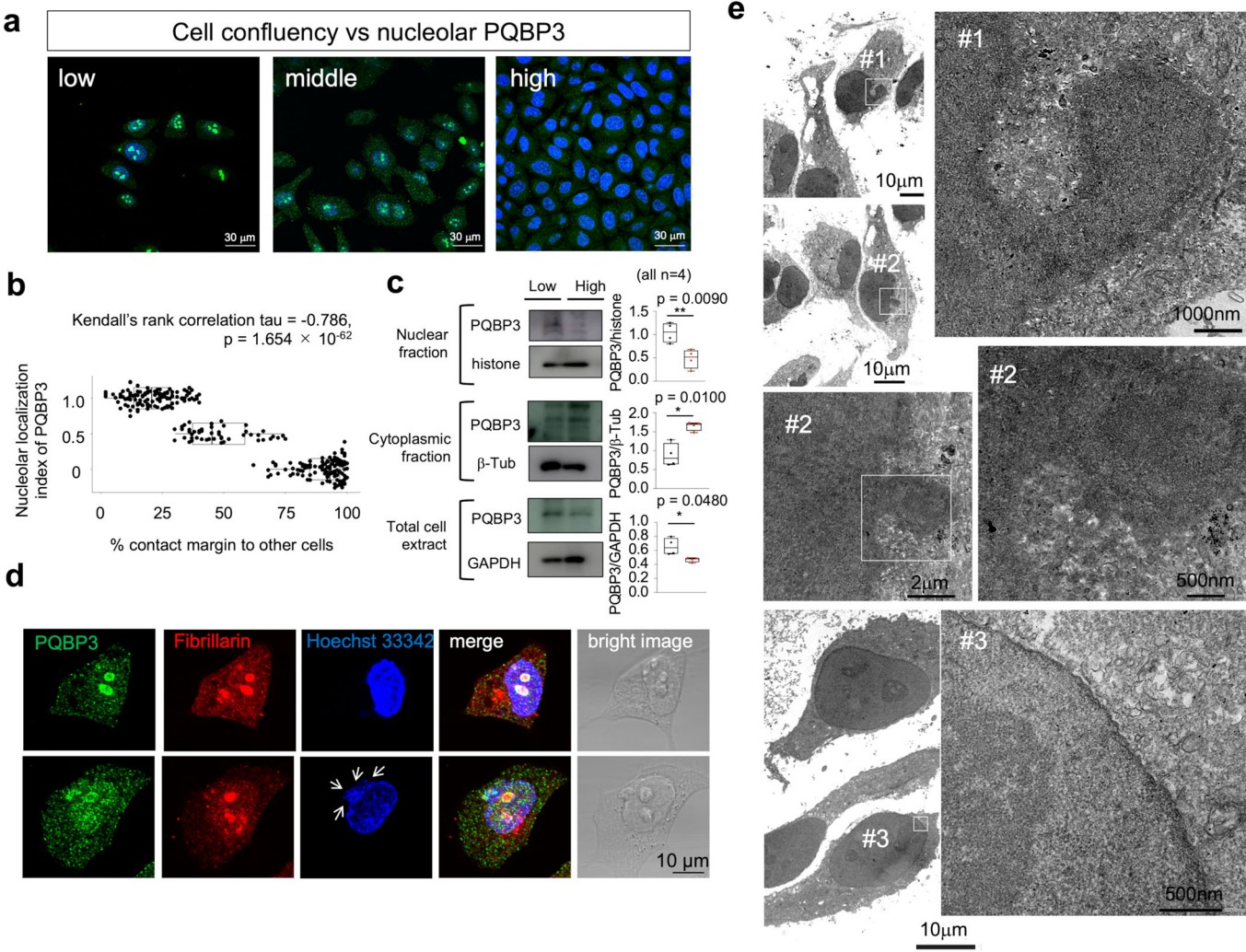

**Figure 2. Nucleolar PQBP3/NOL7 is decreased in quiescence induced by cell-cell contact inhibition.**

(A) PQBP3/NOL7 immunocytochemistry of HeLa cells at various cell densities. High signal intensities of PQBP3/NOL7 were observed in cells at low cell densities, while the signals were dispersed into the cytosol in cells at medium densities and eliminated in confluent cells (high density). (B) Relationship between percent cell margin in contact with neighboring cells and nucleolar distribution of PQBP3/NOL7. Localization of PQBP3/NOL7 was semiquantitated into three stages weighed by different values (1: robust nucleolar PQBP3/NOL7, 0.5: modest nucleolar PQBP3/NOL7, 0: no nucleolar PQBP3/NOL7). A Kendall's rank correlation test revealed a negative relationship between % cell contact margin and PQBP3/NOL7 nucleolar distribution (tau = $-0.786$, $p = 1.654 \times 10^{-62}$). (C) Western blot of PQBP3/NOL7 with nuclear fraction, cytoplasmic fraction and total cell extract of HeLa cells cultured at low and high cell densities. Box plots show the median and 25–75th percentile, and whiskers represent data outside the 25–75th percentile range. (D) Representative images of cells in the stages of "robust nucleolar PQBP3/NOL7" (upper panels) and "faint nucleolar PQBP3/NOL7" (lower panels). In cells classified as "faint nucleolar PQBP3/NOL7," nuclear PQBP3 signals became obscure, though fibrillarin signals of nucleoli were robust, and abnormal protrusion of nuclear margin and extranuclear DNA stains were observed in Hoechst 33342 labeling (white arrows). (E) Electron microscopy of HeLa cells with faint nucleolar PQBP3/NOL7 (#1, #2) and robust nucleolar PQBP3/NOL7 (#3). Nuclear membrane of #1 and #2 cells became obscure and their protrusion contains chromatin. Nucleoli of #1 and #2 cells did not show normal substructures. The continuity of the protrusion and the nucleus excluded that such protrusions were micronuclei. Cells with robust nucleolar PQBP3/NOL7 (#3) showed normal structures of nucleoli and nuclear membrane. Experiments in this figure were technically replicated until the necessary N was acquired. Source data are available online for this figure.

senescence (Lee et al, 2006), confirmed senescence of HeLa cells after ten passages (Fig. 3A,B). In addition, HeLa cells from >10G to >20G revealed increase of cytoplasmic DNA (Fig. 3A). In this senescence condition, we confirmed that PQBP3/NOL7 was shifted from nucleus to cytoplasm while the total amount of PQBP3/NOL7 in senescent HeLa cells was not changed remarkably (Fig. 3C), and revealed the increase of cytoplasmic DNA by subtracting total signal intensity of Hoechst 33342 in nucleus from that in total cell (Fig. 3D). Consistently, western blot revealed decrease of PQBP3/

NOL7 in nuclear fraction and increase of PQBP3/NOL7 in cytoplasmic fraction (Fig. 3E). Total PQBP3/NOL7 in >10G cells was not changed while slightly decreased in >20G cells (Fig. 3E).

## MTOR contributes to cytoplasmic shift of PQBP3/NOL7

In cultured immortalized cells, confluency suppresses cell proliferation via contact inhibition but does not lead to senescence as cells resume proliferation after replating (Leontieva et al, 2014).

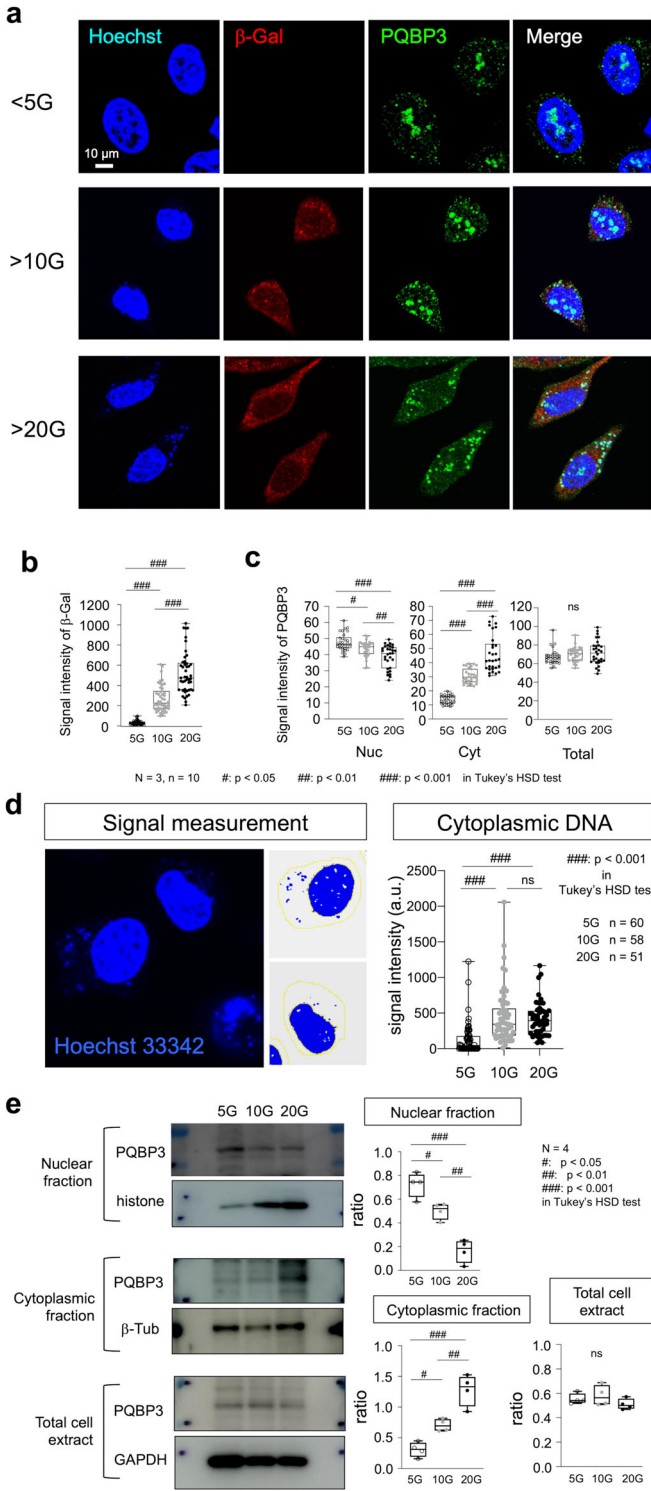

**a**

| | Hoechst | β-Gal | PQBP3 | Merge |
|---|---|---|---|---|
| <5G | | | | |
| >10G | | | | |
| >20G | | | | |

10 μm

**b**

Signal intensity of β-Gal

### ###
### ###

5G 10G 20G

N = 3, n = 10    #: p < 0.05  ##: p < 0.01  ###: p < 0.001  in Tukey's HSD test

**c**

Signal intensity of PQBP3

Nuc / Cyt / Total

**d**

Signal measurement | Cytoplasmic DNA

Hoechst 33342

signal intensity (a.u.)

### ns
###

5G 10G 20G

###: p < 0.001 in Tukey's HSD test

5G  n = 60
10G  n = 58
20G  n = 51

**e**

5G 10G 20G

Nuclear fraction: PQBP3 / histone
Cytoplasmic fraction: PQBP3 / β-Tub
Total cell extract: PQBP3 / GAPDH

Nuclear fraction

ratio

5G 10G 20G

N = 4
#:  p < 0.05
##:  p < 0.01
###: p < 0.001
in Tukey's HSD test

Cytoplasmic fraction

Total cell extract

ratio  ratio

5G 10G 20G   5G 10G 20G

**Figure 3. Nucleolar PQBP3/NOL7 is decreased and shifted to cytoplasm in senescence.**

(A) PQBP3/NOL7 immunocytochemistry of HeLa cells at less than 5 passages (<5G), more than 10 passages (>10G), and more than 20 passages (>20G). Signals were detected in β-Gal staining of >10G and >20G cells in which PQBP3/NOL7 was decreased in the nucleus and shifted to the cytoplasm. (B) Signal intensities of β-Gal were quantified in HeLa cells (30 cells from 3 wells) and compared among three groups. Statistical significance was found in comparison of <5G and >10G (###: $p < 0.0001$), <5G and >20G (###: $p < 0.0001$), and >10G and >20G (###: $p < 0.0001$). (C) Signal intensities of PQBP3/NOL7 in nucleus, cytoplasm, or total cell were quantified and compared among three groups (30 cells from 3 wells). (Nuc) Statistical significance was found in comparison of <5G and >10G (#: $p = 0.027$), <5G and >20G (###: $p < 0.0001$), and >10G and >20G (##: $p = 0.0051$). (Cyt) Statistical significance was found in comparison of <5G and >10G (###: $p < 0.0001$), <5G and >20G (###: $p < 0.0001$), and >10G and >20G (###: $p < 0.0001$). (D) Schematic presentation of the method to quantify the extranuclear DNA signals stained by Hoechst 33342 (left panel). The signal intensities were compared among three groups (right graph). Statistical significance was found in comparison of <5G and >10G (###: $p < 0.0001$), and <5G and >20G (###: $p < 0.0001$). (E) Western blot analyses of nuclear, cytoplasmic, and total PQBP3/NOL7 in 5G, >10G, and >20G HeLa cells (left panels). Statistical comparisons of band intensities among the three groups (right graphs). (Nuclear fraction) Statistical significance was found in comparison of <5G and >10G (#: $p = 0.0189$), <5G and >20G (###: $p < 0.0001$), and >10G and >20G (##: $p = 0.0015$). (Cytoplasmic fraction) Statistical significance was found in comparison of <5G and >10G (#: $p = 0.025$), <5G and >20G (###: $p < 0.0001$), and >10G and >20G (##: $p = 0.003$). Experiments in this figure were technically replicated until the necessary N was acquired. Box plots show the median and 25–75th percentile, and whiskers represent data outside the 25–75th percentile range. Source data are available online for this figure.

mTOR activation shifts cells from quiescence to senescence (Leontieva et al, 2014).

Based on the knowledge, we hypothesized that mTOR activity is related to the cytoplasmic shift of PQBP3/NOL7. We employed MHY1485, a cell-permeable mTOR activator (Fig. 4A), and revealed that mTOR activation enhanced the cytoplasmic shift with western blot (Fig. 4B,C) and with immunocytochemistry (Fig. 4D,E).

## PQBP3/NOL7 knockdown induces nuclear membrane vulnerability and cytoplasmic DNA leakage

A previous study suggested that PQBP3/NOL7 knockdown did not significantly affect nucleolus or cell structure in H1299 human lung carcinoma cells (Kinor and Shav-Tal, 2011), while they over-expressed PQBP3/NOL7-YFP and subsequently knocked down the protein by siRNA (Kinor and Shav-Tal, 2011). We decided to re-examine the effect of PQBP3/NOL7 on nucleolus simply by knocking down endogenous PQBP3/NOL7. In addition, we investigated how knockdown of endogenous PQBP3/NOL7 affected cell morphology and cytoplasmic DNA, and whether the phenotype was relevant to senescence (Fig. 5).

Compared to nontransfected cells, HeLa cells transfected with two types of PQBP3 siRNA exhibited robust decrease of PQBP3/NOL7 in western blot analysis (Fig. 5A) and in immunostaining (Fig. 5B). A population of PQBP3-siRNA-transfected cells exhibited an irregularly shaped nucleus characterized by a notch (Fig. 5B, upper panels, arrow) and uneven distribution of nuclear chromatin (Fig. 5B). In addition, a part of PQBP3-siRNA-transfected cells revealed abnormal shape of nuclei (Fig. 5C), as we observed in a

The mechanism distinguishing reversible quiescence-like contact inhibition and irreversible senescence is thought to be regulated by the activity of mammalian target of rapamycin (mTOR), which phosphorylates p70 S6 kinase (S6K) and eukaryotic translational initiation factor eIF4E-binding protein 1 (4E-BP1) to mediate eIF4E-mediated translation (Leontieva et al, 2014; Sun, 2014). mTOR activity is higher in senescence than in quiescence, and

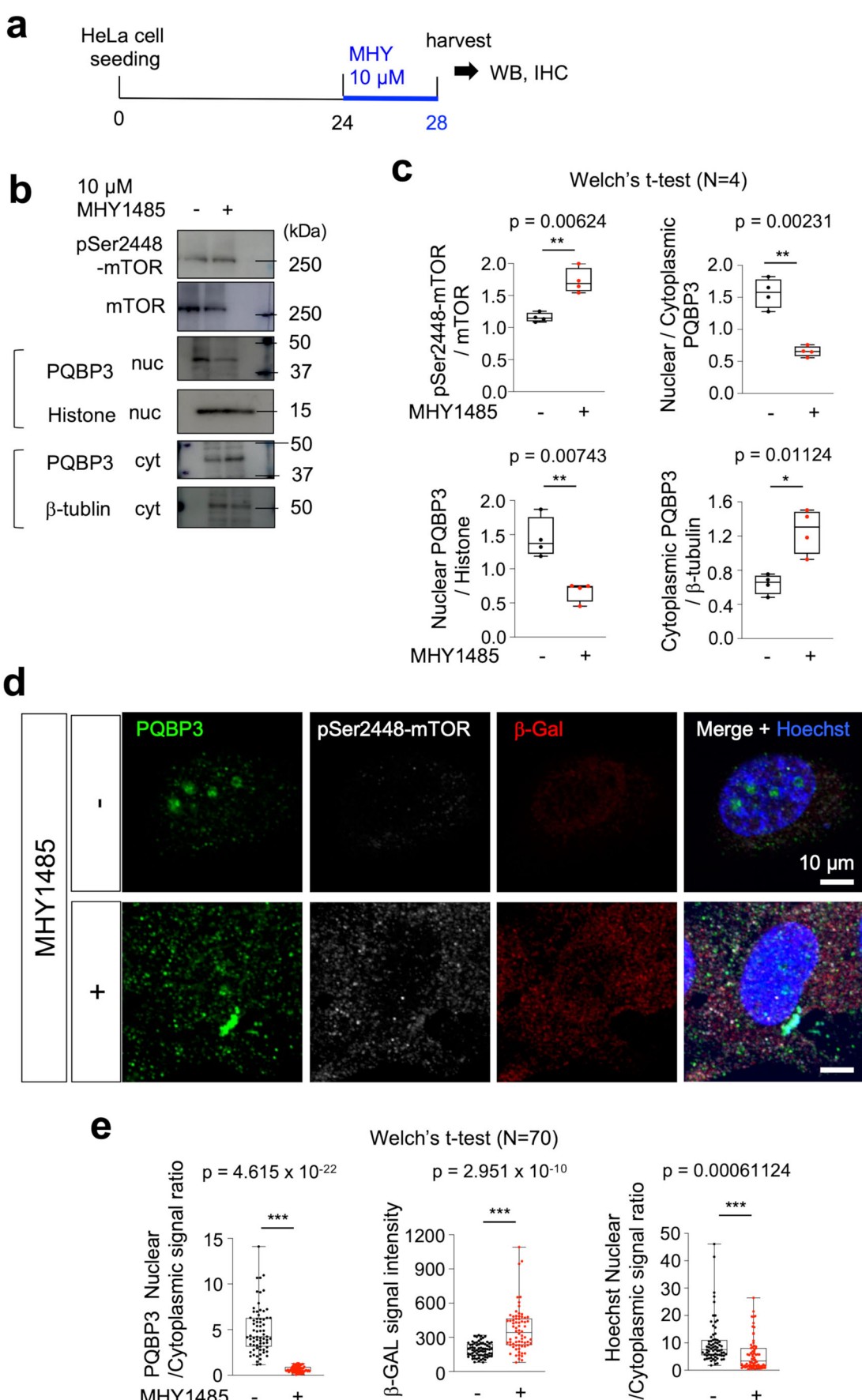

◀ **Figure 4.  MTOR contributes to cytoplasmic shift of PQBP3/NOL7.**

(A) protocol of mTOR signal activation in HeLa cell culture (<5G). MHY1485, an activator of mTOR (final concentration in the medium: 10 µM) was added to the culture medium 48 h after cell seeding and cells were harvested after another 4 h. (B) Western blot analysis of nuclear and cytoplasmic PQBP3/NOL7 under mTOR activation. (C) Quantitative analyses of band intensities of western blots. Statistical analyses (Welch's t-test) revealed decrease of nuclear PQBP3/NOL7, increase of cytoplasmic PQBP3/NOL7, and shift of PQBP3/NOL7 from nucleus to cytoplasm. (D) Immunocytochemistry of HeLa cells treated with MHY1485. Cytoplasmic shift of PQBP3/NOL7 and faint signals of β-GAL were detected in HeLa cells treated with MHY1485. (E) Quantitative analyses of cytoplasmic shift of PQBP3 (PQBP3 nuclear/cytoplasmic signal ratio), senescence (β-GAL signal intensity), and cytoplasmic shift of nuclear DNA (Hoechst nuclear/cytoplasmic signal ratio) in HeLa cells by mTOR activation with MHY1485. Experiments in this figure were technically replicated until the necessary N was acquired. Box plots show the median and 25–75th percentile, and whiskers represent data outside the 25–75th percentile range. Source data are available online for this figure.

population of quiescent HeLa cells with faint nucleolar PQBP3/NOL7 staining (Fig. 3A). Quantification of cells with a notched or protruded nucleus confirmed that these phenotypes were associated with PQBP3 knockdown (Fig. 5D).

Moreover, we demonstrated that when we set a similar threshold for signal intensity of Hoechst 33342 in confocal microscopy, PQBP3-siRNA-transfected cells contained cytoplasmic DNA (Fig. 5A, lower panels). Quantification of cytoplasmic DNA signals confirmed that the change was caused by PQBP3 knockdown in HeLa cells (Fig. 5E).

After the stress of lipofection for siRNA knockdown, some cells underwent cell death, while cells in which PQBP3/NOL7 was knocked down by siRNA did not exhibit apoptotic changes (Fig. EV2h–j). The finding was consistent with the notion that apoptosis is inhibited in acutely induced senescent cells (Childs et al, 2014), and supported the relevance of PQBP3/NOL7 to senescence.

## PQBP3/NOL7 knockdown induces senescence

We next asked whether PQBP3/NOL7 knockdown induced changes in chromatin morphology, such as senescence-associated heterochromatic foci (SAHFs) (Narita et al, 2003; Narita, 2007). As the positive control, we treated HeLa cells with hydrogen peroxide and detected nuclear speckles reactive to anti-H3K9me3 antibody (Fig. 6A). PQBP3/NOL7 knockdown by PQBP3-siRNA induced similar changes in regards of β-Gal staining (Fig. 6B) supporting that deficiency of PQBP3/NOL7 induced senescence, even though recent studies indicating that SAHFs are not an indispensable phenotype for senescent cells (Cohn et al, 2023).

Moreover, we used human normal iPSC-derived neurons and confirmed that hydrogen peroxide induced SAHFs by H3K9me3 staining (Fig. 6C). Staining for β-Gal supported that PQBP3/NOL7 knockdown induced senescence also in human iPSC-derived neurons (Fig. 6D). Signals of β-Gal existed in normal neurons as reported previously (Wang et al, 2023; Piechota et al, 2016) but were significantly increased by PQBP3/NOL7 knockdown (Fig. 6D). These results were consistent with previous findings that cells of some aging disorders including Hutchinson–Gilford progeria syndrome (HGPS) showed senescence characteristics and fragile nuclear membrane due to mutations of Lamin A/C, a sub-membrane structural protein (Burtner and Kennedy, 2010).

## PQBP3/NOL7 overexpression suppresses cytoplasmic DNA leakage in senescence-like cells

To investigate the cause-effect relationship between PQBP3/NOL7 and senescence, we constructed plasmids to express two types of EGFP fusion proteins (pEGFP-N1-PQBP3 and pEGFP-C1-PQBP3)

that reproduced nucleolar localization of PQBP3/NOL7 (Fig. EV3), as demonstrated by immunocytochemistry (Fig. 2), and examined whether expression of PQBP3 fusion proteins affected the senescence phenotype (Fig. 7). We transfected pEGFP-N1-PQBP3 and pEGFP-C1-PQBP3 into premature senescent HeLa cells, which had undergone >20 passages (Fig. 3), and in which some cells exhibited cytoplasmic genomic DNA stained by Hoechst 33342 and DAPI, a phenotype characteristic of senescence (Fig. 7A,B).

As expected, cells expressing PQBP3 fusion proteins did not contain cytoplasmic genomic DNA in a fixed state (Fig. 7A) or in a live state (Fig. 7A), suggesting that PQBP3/NOL7 suppressed the senescence-like phenotype of cytoplasmic genomic DNA. Quantitative analysis supported the negative correlation between EGFP-PQBP3 expression and cytoplasmic genomic DNA ($p = 1.087 \times 10^{-7}$, Fisher's exact test), while no relationship was observed between EGFP expression and cytoplasmic genomic DNA (Fig. 7B, lower tables). These findings indicated that PQBP3/NOL7 over-expression suppressed leakage of genomic DNA into the cytoplasm in senescence-like cells.

## PQBP3/NOL7 interacts with the proteasome activator PSME3

To determine the mechanism by which PQBP3/NOL7 suppressed cytoplasmic genomic DNA leakage, we first identified potential PQBP3/NOL7 binding proteins in silico using the PPI database String (version 11.5). Potential candidates that could cause disproportional distribution of chromatin included nuclear membrane proteins such as Lamin B1, Lamin A/C, emerin, MAN1, SUN1/2, LAP2, and BAF, which attach the genome to the nuclear membrane (Wilson and Berk, 2010; Lammerding et al, 2005); nucleoskeleton proteins such as titin, spectrin, actin, and sin anchoring the genome (Simon and Wilson, 2011); and nuclear protein complex cohesin anchoring the genome; and nuclear protein complex cohesin proteins including SMC1, SMC3, RAD21, and STAG1, which maintain the genome compartment (Kim and Yu, 2020). However, these candidates were not predicted to be PQBP3/NOL7 binding proteins (Fig. EV4a). Instead, almost all the predicted PQBP3/NOL7 binding proteins were nucleolar proteins related to rRNA metabolism, except PSME3, a proteasome regulator that forms a doughnut-shaped homoheptamer, which is associated with the proteasome, and ENY2, a factor involved in mRNA export and transcription (https://string-db.org/) (Fig. EV4a). PSME3 and ENY2 could potentially induce nuclear membrane vulnerability and subsequent cytoplasmic DNA leakage. We elected to focus on PSME3 in the present study, as NF-κB (Sun et al, 2016) and stem cell pluripotency (Pecori et al, 2021) are regulated in part by target protein degradation facilitated by

                                                                    

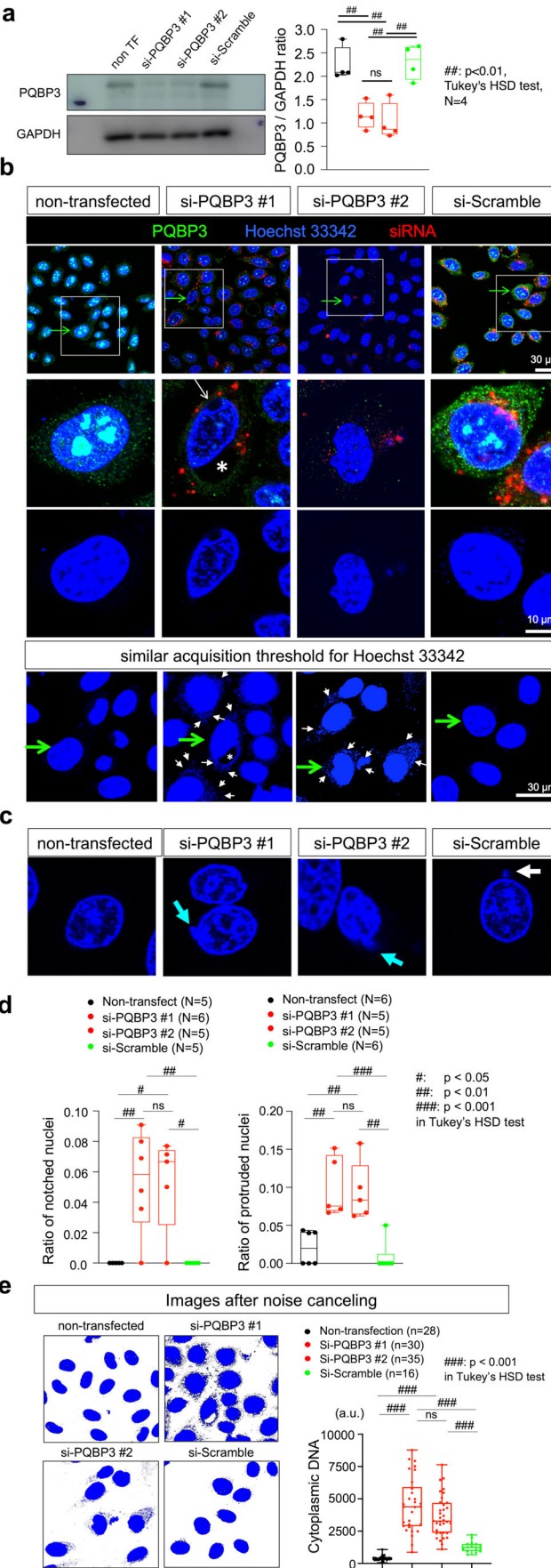

**Figure 5. siRNA PQBP3/NOL7 knockdown induces nuclear morphological abnormalities.**

(A) Western blot analysis of PQBP3/NOL7 in total extracts of HeLa cells transfected with two types of PQBP3-siRNA or scrambled-siRNA. Statistical significance was found in comparison of non TF and si-PQBP3#1 (##: $p = 0.0048$), non TF and si-PQBP3#2 (##: $p = 0.0019$), si-PQBP3#1 and si-Scramble (##: $p = 0.0033$), and si-PQBP3 and si-Scramble (##: $p = 0.0013$). (B) Upper panels show three signals of immunostained PQBP3, Hoechst 33342, and fluorescence-labeled siRNA. Almost all cells were transfected with PQBP3 siRNA (#1, #2), and PQBP3 signals were accordingly reduced. Representative cells from nontransfected, PQBP3 siRNA-transfected (#1, #2), and Scrambled siRNA-transfected cells (green arrow in upper panels) are shown in middle panels. The PQBP3 siRNA-transfected cell exhibited a notched nucleus (white arrow) in which an additional large bleb (asterisk) was formed. When Hoechst 33342 signals were acquired at the same signal intensity threshold, cytoplasmic genomic DNA was present in the PQBP3 siRNA-transfected cell but not in the nontransfected or Scrambled siRNA-transfected cells. (C) Representative Hoechst 33342 images show a normal nucleus in a nontransfected cell (left panel) and an abnormal nucleus with a protrusion (light blue arrows) in a PQBP3-siRNA-transfected cell (middle panel), and a micronucleus (white arrow) that was detected in only a single cell among Scrambled-siRNA-transfected cells (right panel). (D) Quantitative analyses of frequency of cells with notched nuclei (left graph) and frequency of cells with micronuclei (right graph) from 5 to 6 randomly visual fields from independent wells containing 15–50 cells. (left) Statistical significance was found in comparison of non-transfect and si-PQBP3#1 (##: $p = 0.0075$), non-transfect and si-PQBP3#2 (#: $p = 0.0119$), si-PQBP3#1 and si-Scramble (##: $p = 0.0075$), and si-PQBP3#2 and si-Scramble (#: $p = 0.0119$). (right) Statistical significance was found in comparison of non-transfect and si-PQBP3#1 (##: $p = 0.0025$), non-transfect and si-PQBP3#2 (#: $p = 0.0047$), si-PQBP3#1 and si-Scramble (###: $p = 0.0006$), and si-PQBP3#2 and si-Scramble (##: $p = 0.0011$). (E) Quantitative analysis of cytoplasmic DNA signal intensity per cell in nontransfected, PQBP3 siRNA-transfected, and Scrambled siRNA-transfected cells. The original images were corrected by canceling noise signals, and signals outside of the nucleus were measured (see Methods). Cell numbers are shown in the figure, and images were captured from three wells. Statistical significance was found in comparison of non-transfect and si-PQBP3#1 (###: $p = 0.0075$), non-transfect and si-PQBP3#2 (#: $p = 0.0119$), si-PQBP3#1 and si-Scramble (##: $p = 0.0075$), and si-PQBP3#2 and si-Scramble (#: $p = 0.0119$). Experiments in this figure were technically replicated until the necessary N was acquired. Box plots show the median and 25–75th percentile, and whiskers represent data outside the 25–75th percentile range. Source data are available online for this figure.

PSME3 interaction. We thus postulated that a similar mechanism could regulate PQBP3/NOL7 suppression of cytoplasmic DNA leakage. We previously experienced difficulties in the analysis of IDP-IDP interaction by immunoprecipitation. However, PSME3 was not predicted to be an IDP (Fig. EV4b,c), suggesting that immunoprecipitation could effectively determine whether PQBP3/NOL7 and PSME3 interacted.

## PQBP3/NOL7 stabilizes the nuclear membrane via Lamin B1

We postulated that PQBP3/NOL7 could function as a positive or negative regulator of PSME3, a proteasomal activator, affecting degradation of senescence-related proteins such as Lamin B1, which plays critical roles in maintaining the nuclear shape (Shimi et al, 2008), and investigated the relationship between PQBP3/NOL7, PSME3, and LaminB1 using immunocytochemistry and immunoprecipitation (Fig. 8A). Immunocytochemistry using anti-PQBP3, anti-PSME3, and anti-Lamin B1 antibodies revealed that colocalization of PQBP3/NOL7 and PSME3 on the nuclear

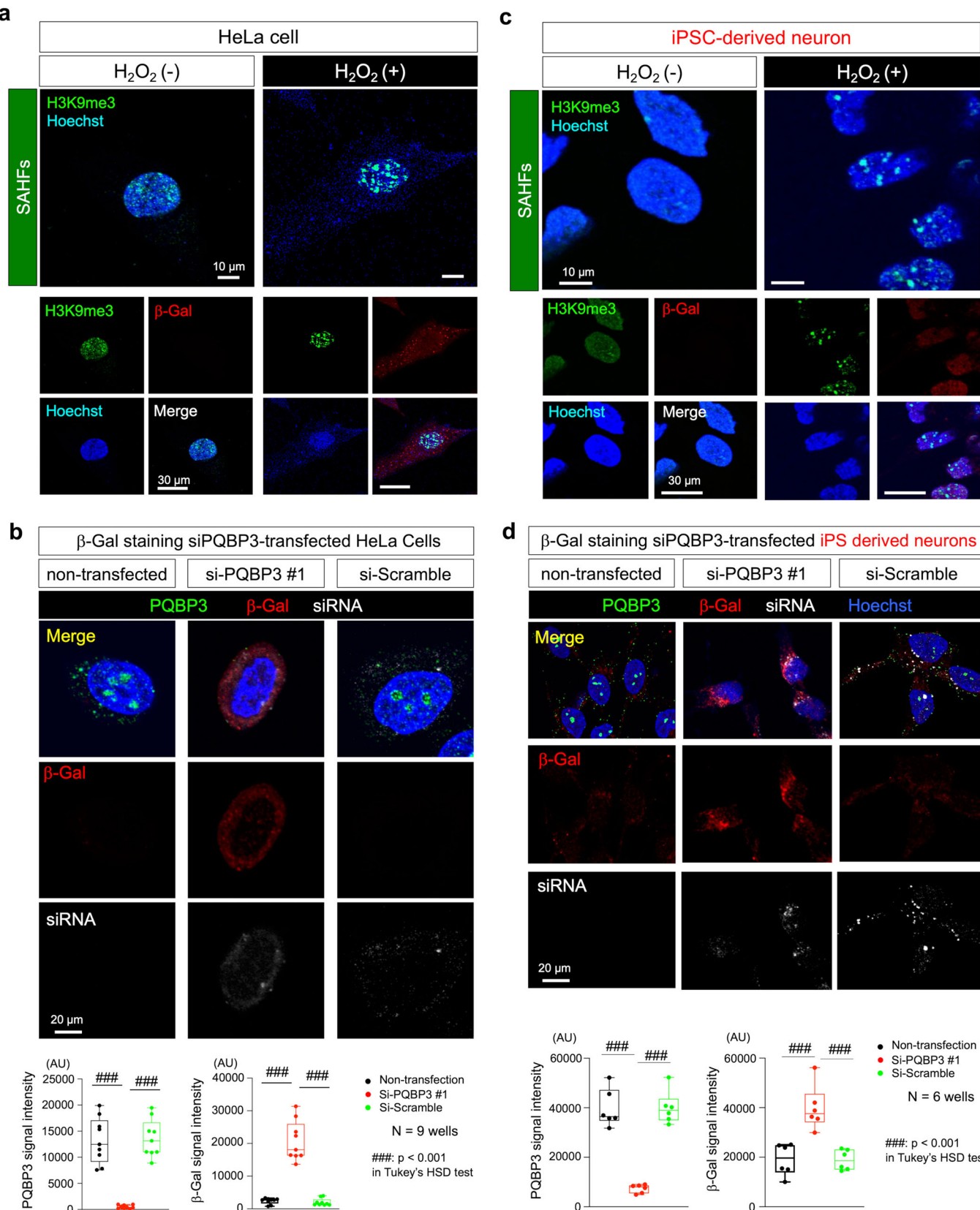

**Figure 6.  PQBP3/NOL7 knockdown induces senescence.**

(A) SAHFs was analyzed in HeLa cells treated with hydrogen peroxide. Nuclear speckles reactive to anti-H3K9me3 antibody indicating SAHFs was observed. Such cells with SAHFs were reactive to β-Gal staining. (B) Positive stains for β-Gal were observed in PQBP3-siRNA-transfected but not Scrambled-siRNA-transfected HeLa cell. (PQBP3) Statistical significance was found in comparison of non-transfection and si-PQBP3#1 (###: $p < 0.0001$), and si-PQBP3#1 and si-Scramble (###: $p < 0.0001$). (β-Gal) Statistical significance was found in comparison of non-transfection and si-PQBP3#1 (###: $p < 0.0001$), and si-PQBP3#1 and si-Scramble (###: $p < 0.0001$). (C) SAHFs was analyzed in human normal iPSC-derived neurons treated with hydrogen peroxide. Nuclear speckles reactive to anti-H3K9me3 antibody was observed similarly to HeLa cells treated with hydrogen peroxide. Such human iPSC-derived neurons with SAHFs were reactive to β-Gal staining. (D) Positive stains for β-Gal were observed in human normal iPSC-derived neurons that were transfected PQBP3-siRNA but not by Scrambled-siRNA-transfected. (PQBP3) Statistical significance was found in comparison of non-transfection and si-PQBP3#1 (###: $p < 0.0001$), and si-PQBP3#1 and si-Scramble (###: $p < 0.0001$). (β-Gal) Statistical significance was found in comparison of non-transfection and si-PQBP3#1 (###: $p < 0.0001$), and si-PQBP3#1 and si-Scramble (###: $p < 0.0001$). Experiments in this figure were technically replicated until the necessary N was acquired. Box plots show the median and 25–75th percentile, and whiskers represent data outside the 25–75th percentile range. Source data are available online for this figure.

membrane was decreased in HeLa cells after hydrogen peroxide induction of senescence (Fig. 8B).

Western blot analysis revealed that senescence induction with hydrogen peroxide decreased PSME3 and Lamin B1 protein levels in HeLa cells transfected with EGFP-PQBP3 or EGFP (Fig. 8C, left panels). Immunoprecipitation with anti-EGFP antibody from such transfected HeLa cells identified a direct interaction between PQBP3/NOL7 and PSME before and after treatment with hydrogen peroxide (Fig. 8C, middle panel). Meanwhile, the interaction between Lamin B1 and PSME3 was decreased by PQBP3/NOL7 expression either in EGFP-PQBP3-transfected or EGFP-transfected HeLa cells transfected (Fig. 8C, middle and lower panels), suggesting that PQBP3/NOL7 suppressed PSME3 interaction with Lamin B1 for ubiquitin-dependent degradation. Consistently, Lamin B1 protein levels in hydrogen peroxide-induced senescence were recovered by PQBP3/NOL7 expression (Fig. 8D).

Interactions between PQBP3, PSME3, and Lamin B1 were further examined by nontransfected HeLa cells with or without treatment of hydrogen peroxide (Fig. 8E). The results from endogenous PQBP3, PSME3, and Lamin B1 were basically consistent with that obtained in HeLa cells overexpressing EGFP-PQBP3 or EGFP. PQBP3 directly interacted with PSME3, while the interaction was suppressed by treatment with hydrogen peroxide (Fig. 8E).

## PQBP3/NOL7 suppresses Lamin B1 ubiquitination

Interestingly, ubiquitinated proteins were decreased overall after induction of senescence, while the band corresponding to ubiquitinated Lamin B1 was increased by PQBP3/NOL7 expression (Fig. 9A). PSME3 is considered to function in ubiquitin-independent pathway of proteasomal degradation (Li and Rechsteiner, 2001; Mao et al, 2008) though its involvement in ubiquitin-dependent proteasomal degradation was also reported (Zhang and Zhang, 2008). In addition, SUMOylation is suggested to mediate such an ubiquitin-independent proteasomal degradation activated by PSME3 (Son et al, 2023). Therefore we examined the identity of the lamin B1 band by immunoprecipitation, and confirmed that precipitated Lamin B1 was ubiquitinated (Fig. 9B). Interestingly, immunoprecipitation results suggested that mono-ubiquitinated Lamin B1 was also SUMOylated (Fig. 9B). We further examined ubiquitination and SUMOylation of Lamin B1 by incubating GST-Lamin B1 for 30 min in lysates prepared from HeLa cells transfected with EGFP-PQBP3 or EGFP-expressing plasmids then treated by $H_2O_2$ (Fig. 9C). Reprobing the same filter with anti-GST, anti-Lamin B1 and anti-SUMO1 antibodies reconfirmed that GST-

Lamin B1 was ubiquitinated and a part of ubiquitinated GST-Lamin B1 was thereafter SUMOylated (Fig. 9C). To confirm the role of PSME3 in Lamin B1 protein degradation, we performed transfection of PSME3-siRNA (KD) and PSME3-expression vector (OE) to HeLa cells, and revealed a negative relationship between PSME3 and Lamin B1, further supporting the involvement of PSME3 in Lamin B1 degradation (Fig. 9D).

Our results revealed simultaneous SUMOylation and ubiquitination of Lamin B1 (Fig. 9B,C), and its reduction in PSME3-OE (Fig. 9D). To differentiate the causative effect of ubiquitination and SUMOylation on PSME3-activated protein degradation of Lamin B1, we performed inhibition experiments of ubiquitination and SUMOylation with TAK-243 and 2-D08 or TAK-981, respectively (Fig. 9E). Inhibition of SUMOylation increased the amount of Lamin B1 protein under PSME3-OE while the effect was not so remarkable by inhibition of ubiquitination, suggesting that the PSME3-activated proteasomal degradation targeted SUMOylated rather than ubiquitinated Lamin B1 (Fig. 9E). The combined immunocytochemistry and immunoprecipitation findings (Figs. 8 and 9) indicated that PQBP3/NOL7 suppresses PSME3-mediated proteasomal degradation of Lamin B1, and that this interaction is inhibited when PQBP3/NOL7 is diminished by induction of senescence.

Prior studies have demonstrated changes to the ubiquitin-proteasome system during senescence (Deschênes-Simard et al, 2014; Fukuura et al, 2019; Marfella et al, 2008; Ullah et al, 2020), and total protein ubiquitination is decreased during aging in lower animals such as C. elegans (Koyuncu et al, 2021), consistent with our findings. Our findings suggested a specific role for PQBP3/NOL7 in suppressing the activity of PSME3, especially for Lamin B1 protein degradation (Fig. EV5).

## Association of PQBP3/NOL7 with the senescence phenotype in polyQ disease pathology

In addition to the effect of PQBP3/NOL7 on senescence, we returned to the originally identified function of PQBP3 to interact with polyQ proteins (Imafuku et al, 1998), investigating the role of PQBP3/NOL7 in polyQ diseases. In our prior study of the nucleolar protein PQBP5/NOL10 (Jin et al, 2023) we revealed that PQBP5/NOL10 was sequestered to polyQ inclusion bodies and depleted from the nucleolus, the functional site of PQBP5/NOL10 (Jin et al, 2023). To determine if similar sequestration of PQBP3/NOL7 to polyQ inclusion bodies occurred, we examined intracellular localization patterns of polyQ disease proteins and PQBP3/NOL7 in their coexpression (Fig. 10A,B). In this case, coexpression of two

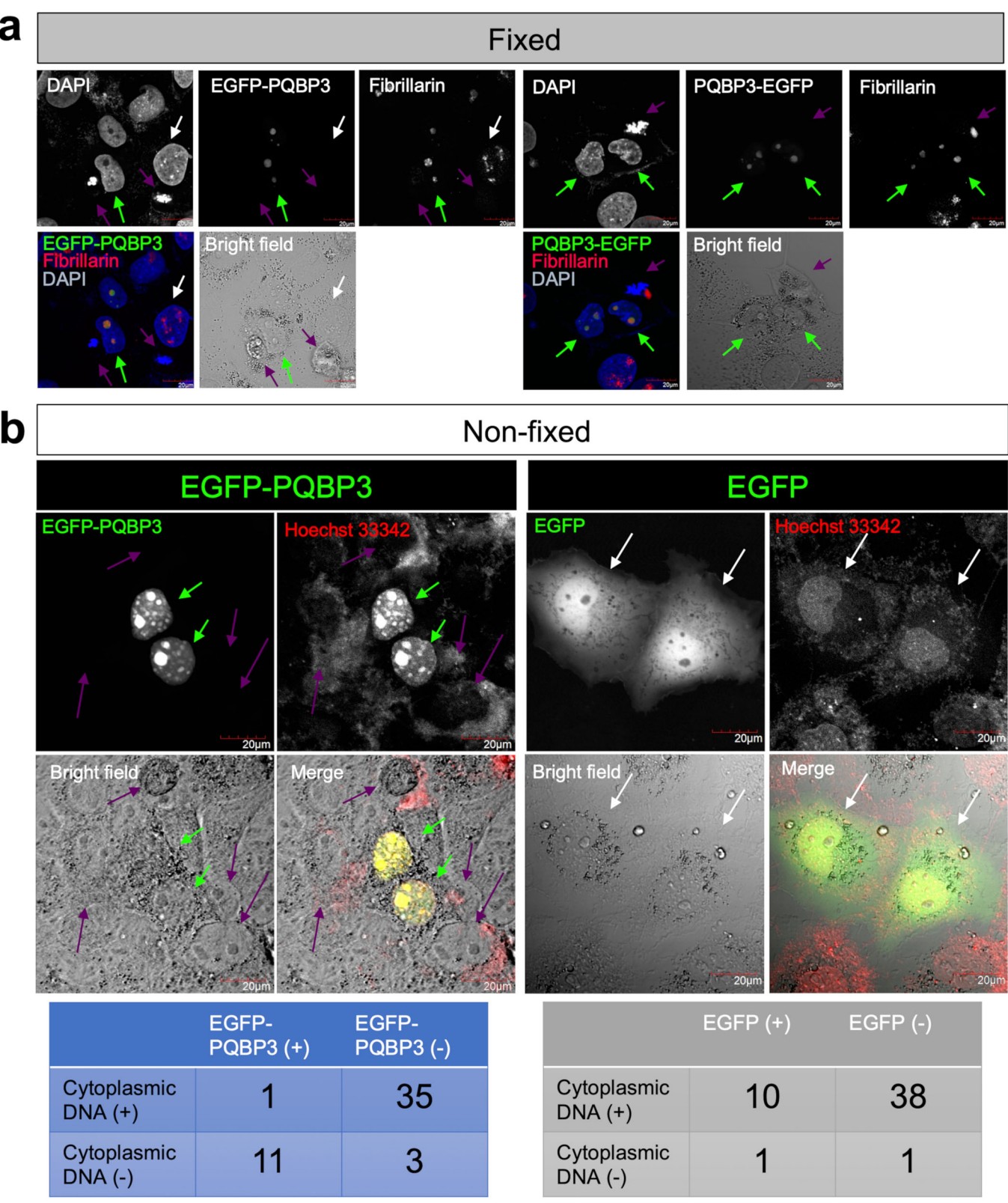

**Figure 7. PQBP3/NOL7 overexpression suppresses cytoplasmic genomic DNA.**

(A) Senescent HeLa cells were fixed with 10% formaldehyde and stained with antifibrillarin and DAPI. Signals of EGFP-PQBP3 fusion proteins were directly detected. Most of the nontransfected cells contained cytoplasmic genomic DNA, as revealed by DAPI (purple arrows), while a small part of nontransfected cells showed normal morphology (white arrows). Transfected cells did not contain cytoplasmic genomic DNA (green arrows). (B) Nonfixed HeLa cells were directly observed to evaluate the effect of EGFP-PQBP3 or EGFP expression on cytoplasmic genomic DNA. In the left panels, nontransfected cells (purple arrows) but not EGFP-PQBP3-expressing cells (green arrows) contained cytoplasmic genomic DNA. In the right panels, cytoplasmic DNA was not affected in EGFP-expressing cells (white arrows) in comparison to neighboring nontransfected cells. Lower tables show quantitative analysis of 50 cells from 5 wells, each transfected with pEGFP-C1-PQBP3 or pEGFP-C1 plasmid. A strong negative correlation between EGFP-PQBP3 expression and cytoplasmic genomic DNA was statistically confirmed by Fisher's exact test ($p = 1.087 \times 10^{-7}$, $n = 5$ wells, $n = 50$ cells), while no relationship was detected between EGFP expression and cytoplasmic genomic DNA. Experiments in this figure were technically replicated until the necessary N was acquired. Source data are available online for this figure.

proteins was preferred over observation of endogenous PQBP3/NOL7 under single expression of a polyQ disease protein, as we had previously determined that expression of polyQ proteins primarily affects the nucleolar structure via PQBP5/NOL10 (Jin et al, 2023). Therefore, coexpression of polyQ and PQBP3/NOL7 fluorescent fusion proteins would more sensitively and specifically detect the effect of polyQ disease proteins on PQBP3/NOL7.

We transiently expressed wild-type and mutant forms of polyQ disease proteins, Ataxin-1 (Atxn1, the causative gene product for spinocerebellar ataxia type-1, SCA1), Ataxin-7 (Atxn7, the causative gene product for spinocerebellar ataxia type-7, SCA7), huntingtin (Htt, the causative gene product for Huntington's disease, HD), and androgen receptor (AR, the causative gene product for spinal bulbar muscular atrophy/Kennedy's disease, SBMA/KD), with or without EGFP-PQBP3, in HeLa cells (Fig. 10A). Unexpectedly, when EGFP-PQBP3 was coexpressed with Atxn1, Htt, and AR, these polyQ disease proteins were sequestered to the PQBP3/NOL7-positive nucleolus, rather than the expected sequestration of PQBP3/NOL7 to polyQ disease protein inclusion bodies (Fig. 10B). These findings potentially suggest that the ability of PQBP3/NOL7 to localize to the nucleolus was more robust than the sequestration effects of Atxn1, Htt, and AR, and that any potential functional impairment of PQBP3/NOL7, in SCA7, HD, or SBMA/KD would not be based on sequestration to inclusion bodies. On the other hand, overexpression of PQBP3/NOL7 shifted both wild-type and mutant forms of Atxn7 from fibrillarin-negative inclusion bodies to nucleoli composed of PQBP3/NOL7 and fibrillarin (Fig. 10B).

Senescence is induced in various conditions. We investigated in this study contact inhibition in confluent cell culture, proliferation senescence after large numbers of cell passage, $H_2O_2$-induced senescence and now mutant Atxn1 expression. Enhanced DNA damage has been reported in SCA1 neurodegeneration models (Qi et al, 2007; Barclay et al, 2014; Ito et al, 2015a, 2015b; Taniguchi et al, 2016), and DNA damage is another condition to induce senescence (d'Adda di Fagagna F, 2008; Schumacher et al, 2021; Pessina et al, 2021). As expected, PQBP3/NOL7 was reduced in nucleoli under expression of mutant Atxn1 (Fig. 10A,B) similarly to other conditions. To confirm that all these conditions induced a similar senescence in regards of cell cycle phase, we employed the nuclear to cytoplasmic ratio of DNA helicase B (DHB) that is often used as an indication of G0-as DHB is nuclear during quiescence (Spencer et al, 2013). The sensor DHB-Ven in which amino acids 994–1087 of human DNA helicase B was fused to the yellow fluorescent protein mVenus (Spencer et al, 2013) enabled us to observe the nuclear to cytoplasmic ratio of DHB in HeLa (Fig. 10C), which reflects CDK2 activity and cell cycle phase (Spencer et al,

2013). The assay revealed that confluence senescence, proliferation senescence, $H_2O_2$-induced senescence and mutant Atxn1-induced senescence were basically similar in regards of their cell cycle stage remaining at G0 (Fig. 10C).

Therefore, our results have shown the similar change of PQBP3 not only in replication senescence but also in damage-induced senescence. Since this study aims to understand neuronal senescence in disease it is important to know whether the same molecular interactions could occur in DNA damage-induced neuronal senescence in vivo and not only replicative senescence in vitro. Finally, for this purpose, we examined the relationship between PQBP3/NOL7 and Atxn1 in mutant Atxn1 knock in (Atxn1-KI = Sca1$^{154Q/2Q}$) mice at 9 weeks of age, when onset of the motor dysfunction occurs (Fig. 10D). This allowed us to determine the in vivo effect of normal expression levels of a mutant polyQ protein on endogenous PQBP3/NOL7 expression and localization. Immunohistochemistry of cerebellar sections revealed that nucleolar PQBP3/NOL7 signal was decreased in Purkinje cells of Atxn1-KI mice relative to wild-type littermate controls (Fig. 10D). The signal intensity of Lamin B1 at the nuclear membrane was decreased in the Purkinje cells of Atxn1-KI mice, while nuclear membrane Lamin B1 signal was similar between Atxn1-KI and control mice in granule cells (Fig. 10D). Interestingly, higher magnification of Lamin B1 images of Purkinje cells of Atxn1-KI mice (Fig. 10D) revealed blebbing and notches on the nuclear membrane, similar to senescent HeLa cells under PQBP3-KD conditions (Fig. 5). Further, PSME3 was localized to the nuclei of Purkinje cells in control mice (Fig. 10E), but both PSME3 and PQBP3/NOL7 were dispersed in the cytosol with low signal intensities in Purkinje cells of Atxn1-KI mice (Fig. 10E). In addition, small puncta of ubiquitinated proteins were accumulated at the periphery of nucleus and/or in the cytosol of Purkinje cells of Atxn1-KI mice (Fig. 10E). This alteration of expression patterns and levels of PQBP3/NOL7, PSME3 and Lamin B1 was similar to that of $H_2O_2$-induced senescent HeLa cells (Fig. 8). Immunohistochemistry for PQBP3, Atxn1 and Ubiquitin revealed that such PQBP3 puncta were costained with Atxn1 and Ubiquitin antibodies (Fig. 10F). Quantitative analyses of PQBP3/NOL7 signals that were merged or not merged with Atxn1 indicated sequestration of PQBP3/NOL7 to Atxn1 foci in Purkinje cells of Atxn1-KI mice (Fig. 10F).

## Discussion

In the present study, we determined the molecular function of PQBP3/NOL7, a protein that binds the polyQ tract sequence, revealing its role as a negative regular of PSME3. In this context, PQBP3/NOL7 prevents PSME3-regulated proteasomal degradation

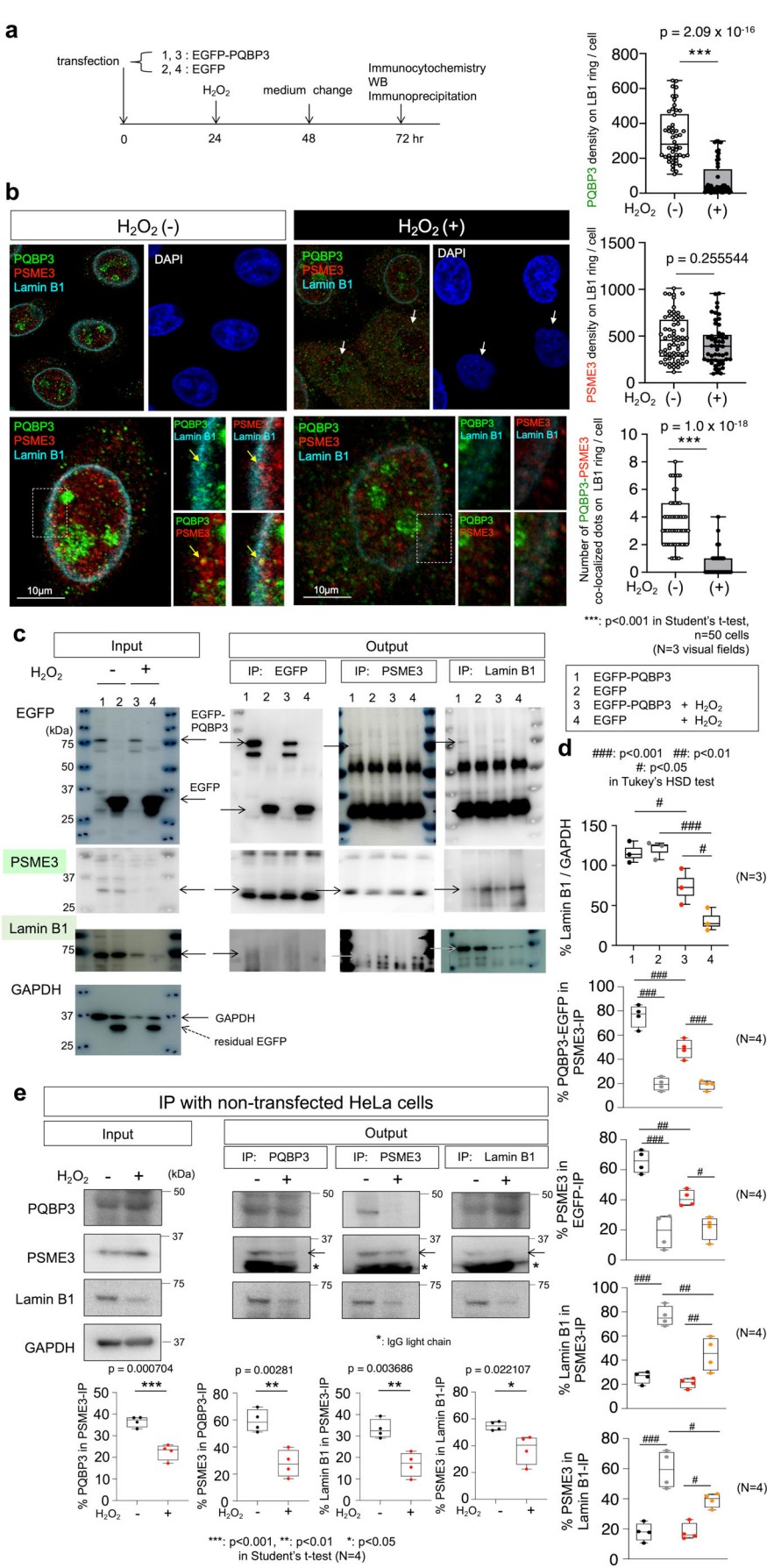

**Figure 8. The tripartite complex of PQBP3/NOL7 and PSME3 stabilizes Lamin B1.**

(A) Protocol for transfection of EGFP-PQBP3 or EGFP expression plasmids and induction of senescence in HeLa cells with hydrogen peroxide ($H_2O_2$). (B) Immunocytochemistry of PQBP3, PSME3, and Lamin B1 in HeLa cells with or without treatment. Upper image panels show HeLa cells with or without $H_2O_2$ treatment. A portion of $H_2O_2$-treated cells (white arrows) lost the Lamin B1 ring at the nuclear membrane, and PQBP3 and PSME3 were dispersed to the cytosol in these cells. The right graphs show signal densities of PQBP3 or PSME3 on the Lamin B1 ring (upper and middle graphs) and numbers of PQBP3-PSME3 colocalized dots on the Lamin B1 ring (lower graph). The signal intensities of PQBP3 and PSME3 on the Lamin B1-positive area were measured by ImageJ. (C) Immunoprecipitation analysis of the interactions between PQBP3, PSME3, and Lamin B1. Left panels and right panels show input and output, respectively, of immunoprecipitations. Input panels show decreased PSME3 and Lamin B1, after $H_2O_2$ treatment. Output panels show suppressed interactions between Lamin B1 and PSME3 by EGFP-PQBP3 in coprecipitation. Black and gray arrows indicate each proteins. (D) Qunatitative analyses of Lamin B1 protein levels normalized to GAPDH. (Lamin B1/GAPDH) Statistical significance was found in comparison of (1) and (3) (#: $p = 0.0452$), (2) and (4) (###: $p = 0.0006$), and (3) and (4) (#: $p = 0.0494$). (PQBP3-EGFP) Statistical significance was found in comparison of (1) and (2) (###: $p < 0.0001$), (1) and (3) (###: $p = 0.0005$), and (3) and (4) (###: $p = 0.0002$). (PSME3 in EGFP-IP) Statistical significance was found in comparison of (1) and (2) (###: $p < 0.0001$), (1) and (3) (#: $p = 0.006$), and (3) and (4) (#: $p = 00252$). (Lamin B1 in PSME3-IP) Statistical significance was found in comparison of (1) and (2) (###: $p < 0.0001$), (2) and (4) (##: $p = 0.0012$), and (3) and (4) (##: $p = 0.0093$). (PSME3 in Lamin B1-IP) Statistical significance was found in comparison of (1) and (2) (###: $p < 0.0001$), (2) and (4) (#: $p = 0.0138$), and (3) and (4) (#: $p = 0.0111$). (E) Immunoprecipitation analysis of the interactions between PQBP3, PSME3, and Lamin B1 was performed by these proteins endogenously expressed in HeLa cells. Experiments in this figure were technically replicated until the necessary N was acquired. Box plots show the median and 25–75th percentile, and whiskers represent data outside the 25–75th percentile range. Source data are available online for this figure.

of Lamin B1, which is localized to the nuclear membrane under physiological conditions but is decreased during senescence. In this context, the nucleolus could function as a storage site for antisenescent PQBP3/NOL7, which translocates to the nuclear membrane under senescence stress. Consistently, we identified that nucleolar PQBP3/NOL7 negatively correlated with cytoplasmic genomic DNA, a typical cell senescence phenotype, under quiescent and senescent states. Deficiency of PQBP3/NOL7 caused nuclear membrane instability and subsequent leakage of nuclear DNA into the cytosol. In this context, cytoplasmic genomic DNA activates cGAS-STING signaling (Motwani et al, 2019; Skopelja-Gardner et al, 2022), as does exogenous cytoplasmic DNA derived from pathogens such as bacteria and viruses, to activate transcription of genes that promote senescence-associated secretary phenotypes (SASPs) (Coppé et al, 2010; Decout et al, 2021). Interestingly, in this signaling cascade, PSME3 perpetuates a positive feedback loop that stimulates NF-κB (Sun et al, 2016). On the other hand, our findings revealed that exogenous expression of PQBP3/NOL7 in senescent cells suppressed cytoplasmic genomic DNA accumulation. Collectively, these data indicate that PQBP3/NOL7 functions as a gatekeeper to prevent catastrophic acceleration of senescence.

The mechanism for Lamin B1 degradation in senescence is not fully understood. Autophagic Lamin B1 degradation is thought to occur in oncogene-induced senescence that is induced in diploid fibroblasts expressing BRAF$^{V600E}$ or H-RAS$^{G12V}$ (Lenain et al, 2015), while involvement of the ubiquitin-proteasome system in Lamin B1 degradation has not been sufficiently addressed. A ubiquitin-like modifier peptide, UB$^{KEKS}$, modifies Lamin B1, whereas the protein modification is linked to subcellular localization but not linked to degradation of Lamin B1 (Dubois et al, 2020). Therefore our findings in this study present a scheme for senescence induction/inhibition that a nucleolar protein PQBP5 is mobilized from nucleolus to nuclear membrane to keep that stability and that the deficiency of PQBP5 promotes nuclear membrane instability and cell senescence.

PSME3, the target of PQBP3/NOL7, is a proteasome activator that forms 11S regulator complex to activate 20S core catalytic complex of proteasome (Mao et al, 2008). Since PSME3 lacks the ATPase activity (Li and Rechsteiner, 2001) PSME3 has been considered to mediate ATP/ubiquitin-independent pathway of proteasomal degradation (Mao et al, 2008), while its involvement in ATP/ubiquitin-dependent pathway was also reported (Zhang and

Zhang, 2008). In addition, SUMOylation of target proteins is suggested to activate degradation by PSME3 (Son et al, 2023). We revealed in this study that hydrogen peroxide treatment induced ubiquitination of Lamin B1 and decreased Lamin B1, whereas this does not necessarily mean Lamin B1 degradation is ubiquitin-dependent. We and others previously detected increase of SUMOylated proteins in human SCA1 brains (Ueda et al, 2002; Steffan et al, 2004) in which ubiquitination is also enhanced (Cummings et al, 1999). Hence, ubiquitination and SUMOylation could proceed in parallel in various conditions including aging and neurodegeneration. Interestingly, SUMOylation of Lamin B1 was reported in spinal cord injury recently (Fan et al, 2024), and we found that hydrogen peroxide treatment enhanced both SUMOylation and ubiquitination of Lamin B1. Inhibition of SUMOylation affects Lamin B1 protein degradation more remarkably than that of ubiquitination, suggesting SUMOylated Lamin B1 is a substrate of PSME3-activated proteasomal degradation. However, mechanisms for PSME3-mediated degradation of Lamin B1 should be further investigated in the future.

Decreased nucleolar PQBP3/NOL7 under quiescence/senescence and suppression of the senescence phenotype by PQBP3/NOL7 overexpression raise the question of whether PQBP3/NOL7 is upstream or downstream of senescence pathways. Inflammation, DNA damage, mitochondrial dysfunction, and other factors activate senescence signaling pathways that ultimately result in inhibition of CDK4/6, the critical executer stopping cell cycle (McHugh and Gil, 2018). Senescence-induced depletion of PQBP3/NOL7 increases cytoplasmic genomic DNA, which mimics the DNA damage phenotype but is independent of the DNA damage response (DDR) signal, and further triggers a subsequent cycle of senescence. In this case, PQBP3/NOL7 is downstream of senescence pathway. Meanwhile, if PQBP3/NOL7 is decreased by non-senescence factors, this event triggers senescence. In this case, PQBP3/NOL7 is upstream of senescence pathway.

In the context of neurodegeneration, overexpression of normal and mutant Atxn1 partially sequestered PQBP3/NOL7 to intra-nuclear Atxn1 inclusion bodies (or Atxn1 speckles). In addition, nuclear PQBP3/NOL7 was decreased, and the nuclear membrane exhibited blebbing in the mouse SCA1 model. These findings support the notion that depletion of nucleolar or nuclear PQBP3/NOL7 by interaction with polyQ disease proteins could induce a pathology like that of senescence. As previously demonstrated for

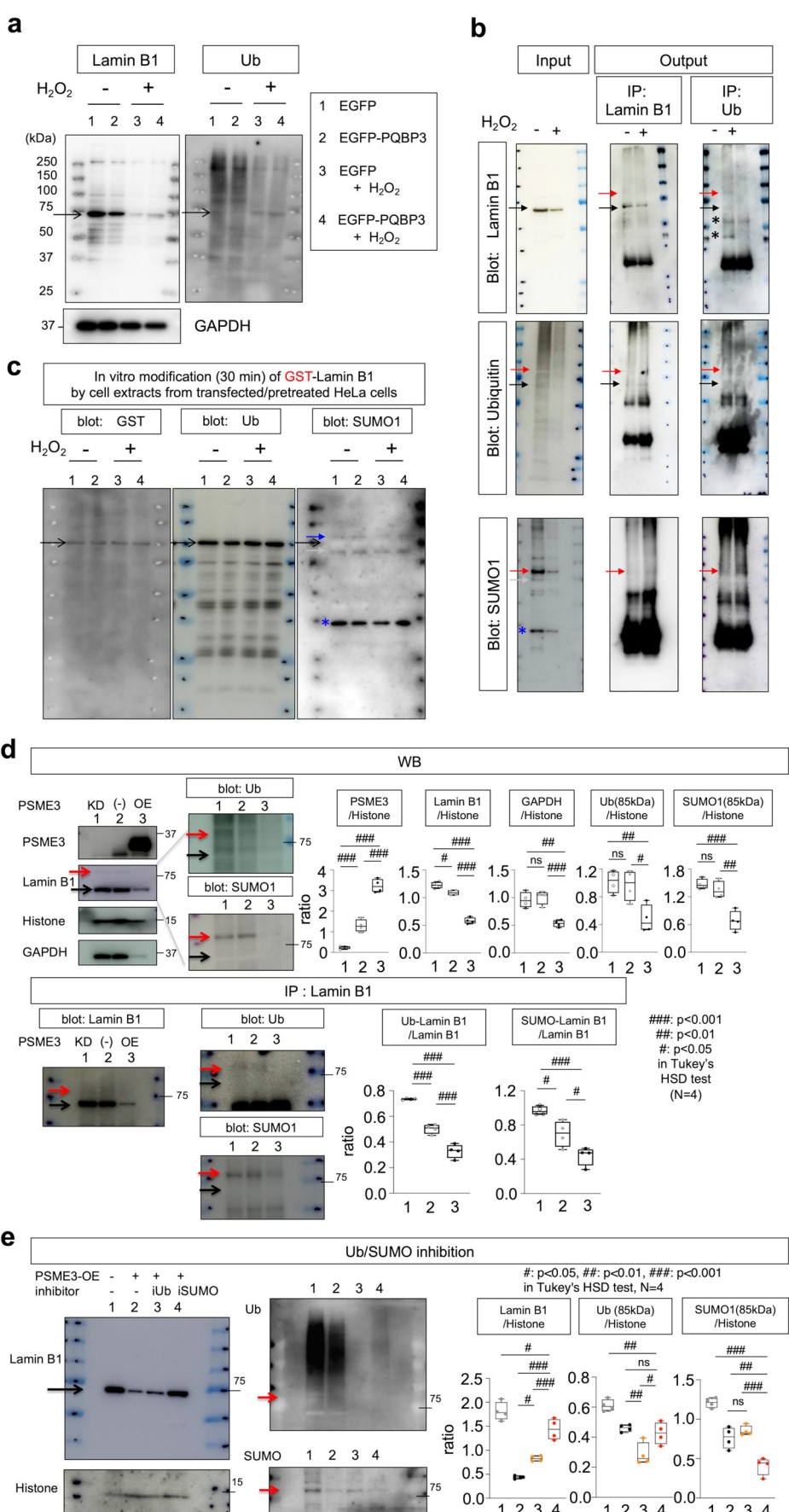

◀ **Figure 9.   Lamin B1 is ubiquitinated and degraded by PSME3.**

(A) Sequential reprobing of the same filter with anti-Ubiquitin antibody and anti-Lamin B1 antibody revealed existence of ubiquitinated Lamin B1 (black arrow), which was increased in HeLa cells treated with hydrogen peroxide despite of the decrease of total Lamin B1. (B) Immunoprecipitation and detection of Lamin B1 by anti-Ubiquitin, anti-SUMO1, and anti-Lamin B1 antibodies to confirm existence of ubiquitinated and SUMOylated Lamin B1. The band detected in (A)) was confirmed by immunoprecipitation as mono-ubiquitinated Lamin B1 (black arrow). In addition, some higher bands were shown as ubiquitinated Lamin B1 in HeLa cells treated with hydrogen peroxide (red arrow). Similarly, SUMOylated Lamin B1 were confirmed. The bands of ubiquitinated Lamin B1 were also reactive to SUMO1 (red arrow), indicating that two modifications occurred simultaneously on Lamin B1. (C) Protein modifications were examined in GST-Lamin B1 and cell lysates from HeLa cells with or without $H_2O_2$ treatment expressing EGFP-PQBP3 or EGFP by transient transfection. Considering the molecular weight 26 kDa of GST, the band indicated with balck arrow corresponds to the band in (A), and it is mono-Ubiquitinated Lamin B1. The band size of blue arrow in SUMO1 blot is consistent with SUMOylated and Ubiquitinated GST-Lamin B1, and corresponds to SUMOylated and Ubiquitinated Lamin B1 detected in immunoprecipitation ((B), red arrow). Light gray arrow-indicated band in (B) corresponds to that in ((C)). (D) PSME3 knockdown by PSME3-siRNA (KD) and PSME3 overexpression by pCMV3-Myc-PSME3 revealed a reverse relationship between PSME3 and Lamin B1. Ubiquitinated and/or SUMOylated LaminB1 was examined by western blot (upper panels) or immunoprecipitation (lower panels). Red and black arrows indicate modified and non-modified Lamin B1 as described above. Right graphs show quantitative analyses of red arrow band intensities in Ub and SUMO blots subtracted by backgrounds and corrected by Lamin B1. (PSME3) Statistical significance was found in comparison of (1) and (2) (###: $p = 0.0007$), (1) and (3) (###: $p < 0.0001$), and (2) and (3) (###: $p < 0.0001$). (Lamin B1) Statistical significance was found in comparison of (1) and (2) (#: $p = 0.0141$), (1) and (3) (###: $p < 0.0001$), and (2) and (3) (###: $p < 0.0001$) (GAPDH) Statistical significance was found in comparison of (1) and (3) (##: $p = 0.0014$), and (2) and (3) (###: $p = 0.0006$). (Ub(85kDa)) Statistical significance was found in comparison of (1) and (3) (##: $p = 0.009$), and (2) and (3) (#: $p = 0.0157$). (SUMO1(85kDa)) Statistical significance was found in comparison of (1) and (3) (###: $p = 0.0004$), and (2) and (3) (##: $p = 0.0012$). (Ub-Lamin B1) Statistical significance was found in comparison of (1) and (2) (###: $p < 0.0001$), (1) and (3) (###: $p < 0.0001$), and (2) and (3) (###: $p = 0.0003$). (SUMO-Lamin B1) Statistical significance was found in comparison of (1) and (2) (#: $p = 0.0182$), (1) and (3) (###: $p = 0.0002$), and (2) and (3) (#: $p = 0.0238$). (E) Inhibitors of ubiquitination (0.1 µM TAK-243) or SUMOylation (10 µM 2-D08) was added to the culture medium of HeLa cells, and the cells were transfected by pCMV3-Myc-PSME3 6 h later. Inhibition of SUMOylation suppressed Lamin B1 decrease by PSME3-OE (black arrow), and the suppressive effect was smaller in inhibition of ubiquitination. Right panels confirm the effects of TAK-243 and 2-D08, respectively, on ubiquitination and SUMOylation, in which the Lamin B1 band reactive to both anti-Ubiquitin and anti-SUMO1 antibodies in Fig. 9B is indicated (red arrows). (Lamin B1) Statistical significance was found in comparison of (1) and (4) (#: $p = 0.0178$), (2) and (3) (#: $p = 0.0126$), (2) and (4) (###: $p < 0.0001$), and (3) and (4) (###: $p = 0.0004$). (Ub) Statistical significance was found in comparison of (1) and (4) (##: $p = 0.003$), (2) and (3) (##: $p = 0.0047$), and (3) and (4) (#: $p = 0.0204$). (SUMO1) Statistical significance was found in comparison of (1) and (4) (###: $p = 0.0002$), (2) and (4) (##: $p = 0.0012$), and (3) and (4) (###: $p = 0.0002$). Experiments in this figure were technically replicated until the necessary N was acquired. Box plots show the median and 25–75th percentile, and whiskers represent data outside the 25–75th percentile range. Source data are available online for this figure.

PQBP1 in the contexts of SCA1, HD, and AD (Waragai, 1999; Okazawa et al, 2002; Tanaka et al, 2018; Jin et al, 2021), PQBP3/NOL7 could function as an additional hub molecule in common pathologies across multiple neurodegenerative diseases. Though the role of PQBP3/NOL7 in other neurodegenerative diseases was beyond the scope of this study, it is important to address how other polyQ proteins affect the function of PQBP3/NOL7.

Multiple human genetics studies (Rass et al, 2007; McKinnon, 2013; Date et al, 2001; Moreira et al, 2001; Takashima et al, 2002; Lee et al, 2015; Bettencourt et al, 2016; Ross and Truant, 2017; Jones et al, 2017) and biological studies of cell/animal models (Homma et al, 2021; Qi et al, 2007; Enokido et al, 2010; Fujita et al, 2013; Madabhushi et al, 2014; Maynard et al, 2015; Taniguchi et al, 2016; Tanaka et al, 2021) of neurodegenerative diseases have demonstrated that neuronal DNA damage is a critical mechanism of neurodegeneration, just like normal human and mouse aging (Lu et al, 2004; Enokido et al, 2008). In these contexts, upstream senescence factors such as DNA damage, senescence-induced depletion of PQBP3/NOL7, and the resultant senescence-like phenotypes such as cytoplasmic DNA leakage and activation of the cGAS-STING pathway (Jin et al, 2021; Mackenzie et al, 2017) comprise a positive feedback loop. Needless to say, multiple factors, which have been identified by various studies (Homma et al, 2021; Qi et al, 2007; Enokido et al, 2010; Fujita et al, 2013; Madabhushi et al, 2014; Maynard et al, 2015; Taniguchi et al, 2016; Tanaka et al, 2021), are also directly or indirectly involved in the feedback loops of neurodegeneration.

Consistent with previous reports that gene mutations or increased protein levels of PQBP3/NOL7 are associated with various cancers (Pinho et al, 2019; Hasina et al, 2006; Doçi et al, 2012; Li et al, 2021), our findings indirectly suggested that PQBP3/NOL7 increases cellular resilience and resistance to cell death in

some contexts. Importantly, the senescence phenotype is induced in some cancer cells, which could affect their responses to anticancer drugs and irradiation (Wang et al, 2022). Given that PQBP3/NOL7 is a gatekeeper of cell aging in senescent or diseased cells, PQBP3/NOL7 is a potential therapeutic target for cancer, neurodegeneration, and aging. However, because the expected effects of therapeutic drugs on cancer and on neurodegeneration or aging are opposite, side effects on the other category of diseases should be rigorously assessed.

An additional finding applicable to basic biology is the relationship between nucleolar proteins and nuclear stability. During senescence, PQBP3/NOL7 is dynamically translocated from the nucleolus to the nucleoplasm, and subsequently from the nucleoplasm to the cytosol. In the present study, we identified that PQBP3/NOL7 suppresses the protein ubiquitination activity of PSME3, preventing Lamin B1 degradation and subsequent nuclear instability. The effect of the PQBP3/NOL7-PSME3 axis on additional target proteins other than Lamin B1 and in other types of neurodegenerative diseases should be investigated in future studies. In addition, phosphorylation and/or other protein modifications of PQBP3/NOL7 could influence its biological activity or cellular localization. Especially, mTOR signaling could affect PQBP3/NOL7, as this axis plays a critical role in senescence (Leontieva et al, 2014). These detailed future mechanistic investigations would delineate the regulatory mechanisms of pathologies such as cancer and neurodegeneration. Interestingly, a recent study performing single-nucleus RNA-seq together with immunohistochemistry of various stages of Alzheimer's disease revealed that tau accumulation is associated with the senescence eigengenes in neurons (Dehkordi et al, 2021). PQBP1, which interact intracellularly with tau to activate the cGAS-STING pathway (Jin et al, 2021), or potentially PQBP3/NOL7, are

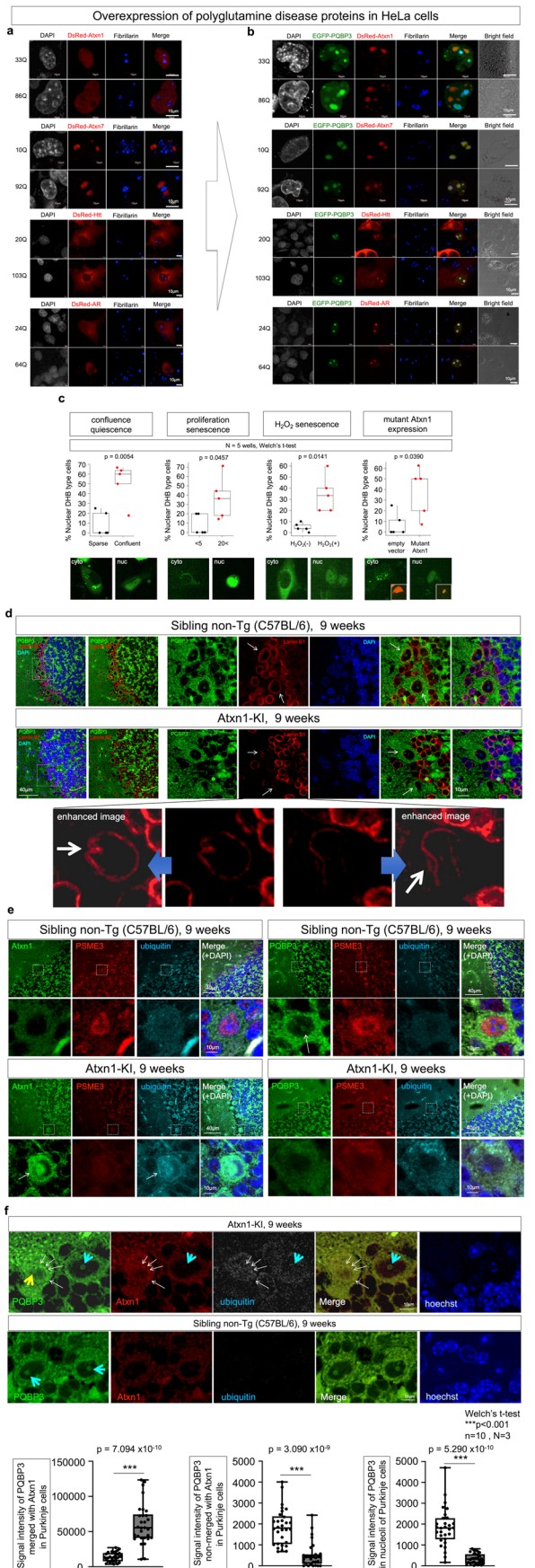

**Figure 10.    PQBP3/NOL7 in cell and animal models of polyQ diseases.**

(A) Colocalization of PQBP3/NOL7 and polyQ disease proteins in cell models. Left panels show single expression of a polyQ disease protein fused to DsRed (DsRed-Atxn1, DsRed-Atxn7, DsRed-Htt, and DsRed-AR) containing a normal or mutant length of polyQ sequence. A small portion of DsRed-Atxn1 formed nuclear speckles in both normal and mutant polyQ lengths, while most of DsRed-Atxn1 exhibited relatively homogeneous nucleoplasm distribution. DsRed-Atxn7 was dominantly distributed in nucleoli. In most cells, DsRed-Htt was predominantly distributed around the nucleus in the cytoplasm, while a part of mutant DsRed-Htt formed cytoplasmic inclusion bodies, as we reported previously (Tagawa et al, 2004). DsRed-AR, in the absence of androgen treatment, was distributed in the cytoplasm. (B) Expression of EGFP-PQBP3 changes the cellular distribution of polyQ disease proteins. EGFP-PQBP3 colocalized with Atxn1 on the nucleoli, while a portion of EGFP-PQBP3 formed nucleoli composed only of PQBP3. EGFP-PQBP3 colocalized with Atxn7 at fibrillarin-positive nucleoleoli. EGFP-PQBP3 coexpression shifted localization of DsRed-Htt and DsRed-AR from the cytoplasm to the nucleoli. (C) Evaluation of senescence by DHB-mVenus in three types of induction, confluence quiescence, proliferation senescence, H2O2 senescence, and mutant Atxn1 expression. Representative nuclear and cytoplasmic images of DHB-mVenus and quantitative analysis of the percentage of senescence-phenotype cells, in which DHB-mVenus was distributed only in nucleus, among total pDHB-mVenus-transfected cells are shown. (D) Immunohistochemistry for PQBP3/NOL7 and Lamin B1 in cerebellar cortex of Atxn1-KI mice and littermate controls at 9 weeks of age. Nucleolar PQBP3/NOL7 signal was decreased in Purkinje cells of Atxn1-KI mice, and nuclear membrane Lamin B1 signal in Purkinje cells also decreased (white arrows). High-magnification original and enhanced images revealed notches and blebbing of nuclear membranes in Purkinje cells of Atxn1-KI mice (white arrow). On the other hand, Lamin B1 signals were unaltered in granule cells. (E) Immunohistochemistry for Atxn1/PSME3/Ubiquitin (left panels) and PQBP3/PSME3/Ubiquitin (right panels) in cerebellar cortexes prepared from Atxn1-KI mice and littermate controls at 9 weeks of age. In a portion of Purkinje cells of Atxn1-KI mice, Atxn1 formed nuclear inclusions with ubiquitin (white arrow), while nuclear PSME3 was decreased. PQBP3 was detected in Purkinje cell nuclei of control mice (white arrow) but decreased in Purkinje cell nuclei of Atxn1-KI mice. PSME3 was localized to the nucleus of Purkinje cells in control mice but dispersed to the cytoplasm in Atxn1-KI mice. These changes in expression and localization of PSME3, PQBP3, and Lamin B1 were homologous to those observed in $H_2O_2$-induced senescent HeLa cells (Fig. 8). (F) Immunohistochemistry for PQBP3/Atxn1/Ubiquitin (left panels) in cerebellar cortexes prepared from Atxn1-KI mice and littermate controls at 9 weeks of age. PQBP3 puncta located at the periphery of nucleus and/or in the cytoplasm (white thin arrows) were costained with Atxn1 and Ubiquitin antibodies in abnormal Purkinje cells (yellow thick arrow), while nucleolus of relatively normal Purkinje cells (light blue thick arrow) was also found in Atxn1-KI mice at 9 weeks. Lower graphs show quantitative analyses of signal intensity of PQBP3 merged with Atxn1 in Purkinje cells, signal intensity of PQBP3 non-merged with Atxn1 in Purkinje cells, and signal intensity of PQBP3 in nucleoli of Purkinje cells. Experiments in this figure were technically replicated until the necessary N was acquired. Box plots show the median and 25–75th percentile, and whiskers represent data outside the 25–75th percentile range. Source data are available online for this figure.

candidate genes that could connect tau accumulation with the senescence phenotype.

Senescence in cancer is controversial. Senescent tumor cells cease proliferation, and therefore senescence is considered to be beneficial for decreasing the rate of tumor mass enlargement and overall malignancy (Wang et al, 2022; Schmitt et al, 2022). On the other hand, senescent tumor cells are protumorigenic in some cases and acquire resistance to cancer therapeutics (Wang et al, 2022; Schmitt et al, 2022), potentially promoting conversion to cancer stem cells (Milanovic et al, 2018). Currently, the effect of cancer-associated mutations on PQBP3/NOL7 functional changes has not been elucidated. Our findings in the present study that PQBP3/NOL7 inhibits senescence could provide a hint for determining

how functional changes of PQBP3/NOL7 regulated by cancer-associated mutations of the human PQBP3/NOL7 gene could influence cancer pathogeneses.

# Methods

## Cell culture

HeLa cells (RCB0007) were cultured in Dulbecco's modified Eagle's medium (Sigma-Aldrich, #5796, MO, USA) containing 10% fetal bovine serum (#15140-122, Gibco, NY, USA) and 1% penicillin-streptomycin (#10270-106, Gibco, NY, USA). Cells were incubated in 5% $CO_2$ at 37 °C. Cells were passaged once every 4 days. To detach cells, after washing with PBS three times, 1 ml 0.25% Trypsin-EDTA (#25200-056, Gibco, NY, USA) diluted with 5x volumes of PBS was added to dishes for 5 min at 37 °C, and culture medium was added to terminate the trypsin reaction prior to cell collection. The cell suspension was then centrifuged, and the supernatant was aspirated. Cell pellets were resuspended in medium and seeded into new culture dishes at a 1:10 ratio.

## Immunocytochemistry

HeLa cells were fixed with 4% formaldehyde and subsequently permeabilized by incubation with 0.1% Triton X-100 in PBS for 10 min at room temperature (RT). After blocking with 0.1% tween 20, 300 mM glycine diluted in PBS containing 10 mg/ml BSA or with PBS containing 10% fetal bovine serum for 30 min at room temperature, cells were incubated with primary antibodies for 16 h at 4 °C and with secondary antibodies for 1 h at RT. The following primary antibodies were used: mouse anti-NOL7 antibody (1:100 or 1:50 for each experiment, #H00051406-B01P, Abnova, Taipei City, Taiwan), rabbit antifibrillarin antibody (1:100, #ab166630, Abcam, Cambridge, UK), and mouse antinucleolin antibody (1:5000, #ab13541, Abcam, Cambridge, UK). The following secondary antibodies were used: antimouse IgG Alexa Fluor 488-conjugated (1:1000, #A21202, Thermo Fisher Scientific, MA, USA), donkey antirabbit IgG Alexa Fluor 594-conjugated (1:1000, #A21206, Thermo Fisher Scientific, MA, USA), and antimouse IgG Alexa Fluor 647-conjugated (1:1000, #A31571, Thermo Fisher Scientific, MA, USA). Nuclei were stained with Hoechst 33343 (1:100, #H342, DOJINDO Laboratories, Kumamoto, Japan) for 30 min at 37 °C or DAPI (1:5000, #D523, DOJINDO Laboratories, Kumamoto, Japan) 15 min at RT. All images were acquired with confocal microscopy (FV1200IX83, Olympus, Tokyo, Japan).

## siRNA knockdown of PQBP3/NOL7

Human NOL7-siRNA (#sc-95562, Santa Cruz, Dallas, TX, USA) or Trilencer-27 universal scrambled negative control siRNA duplex (#SR30004, OriGene, Rockville, MD, USA) was labeled using a Label IT siRNA Tracker Cy5 Kit (#MIR7213, Mirus, WI, USA). One pmol Cy5-labeled siRNA was transfected into HeLa cells with Lipofectamine RNAiMAX (#13778075, Thermo Fisher, Waltham, MA, USA). Twenty-four hours later, cells were stained with Hoechst 33343 (1:100, #H342, DOJINDO Laboratories, Kumamoto, Japan) for 30 min at 37 °C, fixed with 4% formaldehyde in PBS at room temperature for 10 min, treated with 0.1% Triton in PBS for 10 min, and blocked with PBS containing 10% fetal bovine serum for 30 min at RT. Cells were incubated with anti-NOL7 antibody (1:50, FL-257 H00051406-B01P, Novus Biologicals) diluted in dilution buffer (2% fetal bovine serum, 0.1% Triton in PBS) for 10 h at 4 °C, followed by incubation with Donkey antimouse IgG (H+L) Alexa Fluor 488 (1:1000, #A21202 Molecular Probes, Eugene, OR, USA).

## Plasmid construction and transfection

To construct the pEGFP-C1-PQBP3 and pEGFP-N1-PQBP3 plasmids, PQBP3-coding inserts were amplified from PQBP3-pcDNA3.1⁺/C-(K)-DYK (GenScript, New Jersey, USA). The primer sets were used as follows:

(pEGFP-C1-PQBP3) forward: 5′-atgcgaattctatggtgcagctccgaccg-3′, reverse: 3′-atgcgtcgacttacttcttagttttcatctttct-5′

(pEGFP-N1-PQBP3) forward: 5′-atgcgaattctatggtgcagctccgaccg-3′, reverse: 3′-atgcggatcccgcttcttagttttcaatctttctga-5′.

After the amplification products were digested by EcoRI and SalI or EcoRI and BamHI, they were subcloned into pEGFP-C1 plasmid (Addgene, MA, USA) or pEGFP-N1 plasmid (Addgene, MA, USA).

To construct the fusion plasmid of Atxn1-33Q/86Q, Atxn7-10Q/92Q, Htt-20Q/103Q, or AR-24Q/64Q in pDsRed-monomer-C1 were generated as described previously (Fujita et al, 2013). Briefly, to construct plasmids expressing Atxn1-33Q/86Q, Atxn7-10Q/92Q, Htt-20Q/103Q, or AR-24Q/64Q in pDsRed-monomer-C1 plasmid (Clontech, CA, USA) was digested by XhoI and EcoRI, HindIII and SalI, BamHI and XhoI, or XhoI and BamHI.

Hela cells were transfected with 3 μg/ml plasmids using Lipofectamine 2000 transfection reagent (#11668019, Thermo Fisher Scientific, Waltham, MA, USA). After 48 h, cells were fixed with 4% formaldehyde.

## Senescence induction

HeLa cells were transfected with pEGFP-C1-PQBP3 and pEGFP-C1 plasmids, respectively, using Lipofectamine 2000 Reagent (#11668019, Thermo Fisher Scientific, Waltham, MA, USA). After reaching 60% confluency, cells were treated with 150 μM hydrogen peroxide (#081-04215, Wako, Osaka, Japan) diluted in high-glucose DMEM (#11965092, Thermo Fisher Scientific, Waltham, MA, USA) containing 10% FBS (#10437, Gibco, MA, USA). After 24 h, medium was removed, and cells were washed thoroughly with PBS and subsequently cultured in fresh medium (high-glucose DMEM (#11965092, Thermo Fisher Scientific, Waltham, MA, USA) containing 10% FBS (#10437, Gibco, MA, USA).

## Subcellular fractionation

HeLa cells were treated with 10 μM MHY1485 an activator of mTOR for 4h, and collected by a cell scraper, and centrifuged at $1000 \times g$ for 5 min at 4 °C. The pellet was resuspended in 8× vol of lysis buffer (20 mM Hepes pH 7.9, 1 mM dithiothreitol (DTT), 1 mM EDTA, 10% Glycerol, 0.5 mM spermidine, and 0.5% protease inhibitor cocktail (Calbiochem, San Diego, CA, USA) with 0.3% Nonidet P-40), placed on ice for 5 min, and centrifuged at $15,000 \times g$ for 10 min at 4 °C. The supernatant was used as cytoplasmic fraction. The pellet was suspended again with 1× vol of the lysis buffer, added with 2 M KCl, mixed gently, placed on ice

for 30 min, and centrifuged at $100,000 \times g$ for 30 min at 4 °C. The supernatant was used as nuclear fraction for Western Blotting. The following primary antibodies were used: anti-mTOR antibody (1:5000, #66888-1-IG, ProteinTech), anti-Phospho-mTOR (Ser2448) antibody (1:1000, #2971S, Cell Signaling Technology).

## Immunoprecipitation

HeLa cells were transfected with pEGFP-C1-PQBP3 and pEGFP-C1 plasmids, respectively, and senescence was induced with 150 μM hydrogen peroxide. Cells were then harvested and lysed with TNE buffer (10 mM Tris-HCl (pH 7.5), 150 mM NaCl, 1 mM EDTA, 1% Nonidet P-40) containing protease inhibitor cocktail (#539134, Merck Millipore). Lysates were rotated for 30 min at 4 °C and centrifuged at $10,000 \times g$ for 5 min at 4 °C. Each supernatant was incubated with a 50% slurry of Protein-G Sepharose beads (17061801, GE Healthcare) for 2 h at 4 °C, followed by centrifugation at $2000 \times g$ for 2 min at 4 °C. The supernatants were incubated with antibody for 16 h at 4 °C with rotation, followed by addition of 40 μL 50% Protein-G Sepharose and rotation for 2 h at 4 °C. The beads were washed three times with TNE buffer, and 30 μL sample buffer (125 mM Tris-HCl (pH 6.8), 4% (w/v) SDS, 5% (v/v) 2-mercaptoethanol, 10% (v/v) glycerol, and 0.0025% (w/v) bromophenol blue) was added to each sample. Antibodies for immunoprecipitation analysis included rabbit anti-PSME3 antibody (1:100, #PA5-17333, Thermo Fisher Scientific, Waltham, MA, USA), rabbit anti-GFP antibody (1:500, #ab6556, Abcam, Cambridge, UK), and rabbit anti-Lamin B1 antibody (1:300, #ab16048, Abcam, Cambridge, UK). Samples were boiled at 95 °C for 10 min followed by SDS-PAGE and transferred to Immobilon-P polyvinylidene difluoride membranes (Millipore). Antibodies for western blot analysis included rabbit anti-PSME3 antibody (1:3000, #PA5-17333, Thermo Fisher Scientific, Waltham, MA, USA), mouse anti-GFP antibody (1:5000, #sc9996, Santa Cruz Biotechnology, Dallas, TX, USA), rabbit anti-Lamin B1 antibody (1:3000, #ab16048, Abcam, Cambridge, UK), mouse anti-Ubiquitin antibody (1:2500, #3936, Cell Signaling Technology, Danvers, MA, USA), mouse anti-GAPDH antibody (1:6000, #MAB374, Millipore), trueblot anti-rabbit IgG HRP (1:2000, 18-8816-33, Rockland Immunochemicals Inc., PA, USA), and trueblot antimouse IgG HRP (1:2000, 18-8817-33, Rockland Immunochemicals Inc., PA, USA).

HeLa cells in culture with DMEM medium (D5796, Sigma-Aldrich, St. Louis, MO, USA) with 10% Fetal Bovine Serum (#10270106, Gibco, MA, USA) were treated with 150 μM hydrogen peroxide for 24 h, and the medium was exchanged to remove hydrogen peroxide. After 24 h, cells were collected with ice-cold PBS, and cell pellets were lysed by 300 μL RIPA buffer (10 mM Tris-HCL (pH 7.5), 150mM NaCl, 1mM EDTA-2Na, 1% Triton X-100, 0.1% SDS, 0.1% deoxycholate, 1:250 protease inhibitor cocktail (Millipore, #539134)). After centrifugation ($10,000 \times g \times 10$ min), in gentle rotation lysates were pre-incubated with protein G sepharose ((17061801, GE Healthcare, Chicago, IL, USA), 50% slurry in RIPA buffer) for 1 h at 4 °C, incubated with rabbit anti-Lamin B1 antibody (#ab16048, Abcam, Cambridge, UK), anti-PSME3 antibody (#PA5-17333, Thermo Fisher Scientific, Waltham, MA, USA) or anti-GFP antibody (#sc9996, Santa Cruz Biotechnology, Dallas, TX, USA) at 1:300 dilution for 16 h at 4 °C, and  mixed with protein G sepharose (50% slurry in RIPA buffer) for 4 h. After three

times of rinse with RIPA buffer, immunoprecipitates were eluted from protein G sepharose with equal volume of sample buffer.

## iPSC culture and differentiation to pan-neurons

Normal iPSCs (ASE-9203, Applied StemCell, CA, USA) were differentiated to pan-neurons as described previously (Tanaka et al 2021). Briefly, iPSCs were cultured in TeSR-E8 medium (STEMCELL Technologies, BC, Canada) with 10 μM Y27632 (253-00513, Wako, Osaka, Japan). After 24 h, medium was changed to Stem Fit (AK02N, Ajinomoto, Tokyo, Japan) containing 5 μM SB431542 (13031, Cayman Chemical, Ann Arbor, MI, USA), 5 μM CHIR99021 (13122, Cayman Chemical, Ann Arbor, MI, USA), and 5 μM dorsomorphin (044-33751, Wako, Osaka, Japan). After 5 days, iPS cells were dissociated with TrypLE Select (12563-011, Thermo Fisher Scientific, MA, USA). Neurospheres were then cultured in KBM medium (16050100, KHOJIN BIO, Saitama, Japan) with 20 ng/mL Human-FGF-basic (100-18B, Peprotech, London, UK), 10 ng/mL Recombinant Human LIF (NU0013-1, Nacalai, Kyoto, Japan), 10 μM Y27632 (253-00513, Wako, Osaka, Japan), 3 μM CHIR99021 (13122, Cayman Chemical, Ann Arbor, MI, USA), and 2 μM SB431542 (13031, Cayman Chemical, Ann Arbor, MI, USA) for 10 days. Finally, neurospheres were dissociated and seeded onto chambers coated with poly-L-ornithine (P3655, Sigma-Aldrich, St. Louis, MO, USA) and laminin (23016015, Thermo Fisher Scientific, Waltham, MA, USA) to differentiate to pan-neurons. The pan-neurons were cultured in DMEM/F12 (D6421, Sigma-Aldrich, St. Louis, MO, USA) supplemented with 2% B27 (17504044, Thermo Fisher Scientific, Waltham, MA, USA), 1% Glutamax (35050061, Thermo Fisher Scientific, Waltham, MA, USA), and 1% penicillin/streptomycin (15140-122, Thermo Fisher Scientific, Waltham, MA, USA).

After 4 days from neurosphere seeding, pan-neurons were transfected with 1.5 μl of 2 μM PSME3 siRNA (#sc-106344, Santa Cruz) with RNAiMAX (#13778075, Thermo Fisher) or pCMV3-Myc-PSME3 (#HG14682-NM, Sino Biological) by Lipofectamine 2000 transfection reagent (#11668019, Thermo Fisher). After another one day, pan-neurons were fixed with 4% formaldehyde, and stained with anti-H3K9Me3 antibody (1:100, #GTX121677, GeneTex), anti-beta-galactosidase antibody (1:200, #15518-1-AP, Protein Tech) and/or anti-PSME3 antibody (1:200, #14907-1-AP, Protein Tech) for 16 h at 4 °C, and with secondary antibodies (anti-mouse IgG Alexa Fluor 488-conjugated (1:1000, #A21202, Thermo Fisher Scientific, MA, USA), donkey anti-rabbit IgG Alexa Fluor 594-conjugated (1:1000, #A21206, Thermo Fisher Scientific, MA, USA) for 1 h at RT. Images were taken by confocal microscopy (FV1200IX83, Olympus, Tokyo, Japan). Pan-neurons were also stained with mouse anti-NOL7 antibody (1:100 for each experiment, #H00051406-B01P, Abnova, Taipei City, Taiwan), rabbit anti-fibrillarin antibody (1:100, #ab166630, Abcam, Cambridge, UK), and/or mouse anti-nucleolin antibody (1:5000, #ab13541, Abcam, Cambridge, UK), and then with Alexa Fluor 488-conjugated anti-mouse IgG (1:1000, #A21202, Thermo Fisher Scientific, MA, USA), Alexa Fluor 594-conjugated donkey anti-rabbit IgG (1:1000, #A21206, Thermo Fisher Scientific, MA, USA) or Alexa Fluor 647-conjugated anti-mouse IgG (1:1000, #A31571, Thermo Fisher Scientific, MA, USA) for 1 h at RT. Images were acquired with super-resolution microscopy (SpinSR10, Olympus, Tokyo, Japan).

## In vitro Lamin B1 ubiquitination

HeLa cells were plated at $1.5 \times 10^6$ cells/10 cm dish and transfected with pEGFP-C1-PQBP3 and pEGFP-C1 plasmids, respectively, and senescence was subsequently induced with 150 μM hydrogen peroxide. Cells were harvested and lysed with 1000 μl TNE buffer per 10 cm dish (10 mM Tris-HCl (pH 7.5), 150 mM NaCl, 1 mM EDTA, 1% Nonidet P-40) containing protease inhibitor cocktail (#539134, Merck Millipore). Lysates were rotated for 30 min at 4 °C and centrifuged at $10,000 \times g$ for 5 min at 4 °C. Each supernatant was mixed with 30 nM GST-tagged recombinant Lamin B1 protein (#H00004001-P01, Abnova, Taipei City, Taiwan) and rotated for 2 h at 25 °C. The samples were mixed with an equal volume of sample buffer (125 mM Tris-HCl (pH 6.8), 4% (w/v) SDS, 5% (v/v) 2-mercaptoethanol, 10% (v/v) glycerol, and 0.0025% (w/v) bromophenol blue) and boiled at 95 °C for 10 min. Samples were separated by SDS-PAGE and transferred to Immobilon-P polyvinylidene difluoride membranes (Millipore) using semidry transfer. Membranes were blocked by incubation in 5% milk in TBST (10 mM Tris/HCl (pH 8.0), 150 mM NaCl, 0.05% Tween-20), and incubated with mouse anti-GFP antibody (1:5000, #sc9996, Santa Cruz Biotechnology, Dallas, TX, USA); mouse anti-Ubiquitin antibody (1:2500, #3936, Cell Signaling Technology, Danvers, MA, USA); or rabbit anti-GST antibody (1:5000, #sc459, Santa Cruz Biotechnology, Dallas, TX, USA) overnight at 4 °C. Membranes were subsequently incubated with HRP-linked antirabbit IgG (1:3000, #NA934, GE Healthcare) or HRP-linked antimouse IgG (1:3000, #NA931, GE Healthcare) secondary antibodies for 1 h at RT. Proteins were detected with ECL Prime Western Blotting Detection Reagent (RPN2232, GE Healthcare) and a luminescent image analyzer (ImageQuant LAS 500, GE Healthcare).

## Inhibition of ubiquitination and SUMOylation

$5 \times 10^5$ HeLa cells were treated with 0.1 μM ubiquitination inhibitor TAK-243 (#S8341, Selleck, Houston, TX, USA) or 10 μM SUMOylation inhibitor 2-D08 (#S8696, Selleck, Houston, TX, USA) for 6 h before cells were transfected with 3 μg pCMV3-Myc-PSME3 (#HG14682-NM, Sino Biological) using Lipofectamine 2000 transfection reagent (#11668019, Thermo Fisher Scientific, Waltham, MA, USA). After 24 h of transfection, cells were collected with TNE buffer (10 mM Tris-HCl (pH 7.5), 150 mM NaCl, 1 mM EDTA, 1% Nonidet P-40) containing protease inhibitor cocktail (#539134, Merck Millipore), mixed with an equal volume of sample buffer (125 mM Tris-HCl (pH 6.8), 4% (w/v) SDS, 5% (v/v) 2-mercaptoethanol, 10% (v/v) glycerol, and 0.0025% (w/v) bromophenol blue) and used for western blot analysis with anti-Lamin B1 antibody (1:10,000, #ab16048, Abcam, Cambridge, UK) or anti-Histone H4 antibody (1:5000, #ab31830, Abcam, Cambridge, UK) followed by HRP-linked antirabbit IgG (1:3000, #NA934, GE Healthcare) or HRP-linked antimouse IgG (1:3000, #NA931, GE Healthcare), respectively.

## DAPI signal measurement

The signal intensity of DAPI was measured using Fiji (with ImageJ 1.53q, National Institutes of Health, USA) (Schindelin et al, 2012). To exclude noise distributed throughout the visual field, the "Threshold" function in Fiji was applied to the blue channel of each image with the "Auto" threshold. After excluding noise, the region of interest (ROI) was manually specified to cover the nuclei and/or cells. The area of the ROI and mean gray value in the ROI were measured, and the integrated density (IntDen) was calculated by the product of [Area of ROI] x [mean gray value in ROI]. IntDen was used as the intensity value of DAPI signal in the ROI.

## Immunoelectron microscopy

Hela cells ($1 \times 10^4$ Cells in 500 μL) were transfected with 6 nM of Cy5-labeled Human NOL7-siRNA (#sc-95562, Santa Cruz, Dallas, TX, USA) or Trilencer-27 universal scrambled negative control siRNA duplex (#SR30004, OriGene, Rockville, MD, USA). After 24 h, the cells were fixed with fixation buffer (4% paraformaldehyde, 0.1% glutaraldehyde in 0.1 M phosphate buffer) for 15 min at 25 °C and incubated in blocking buffer (50 mM $NH_4Cl$, 0.1% Saponin, 1% BSA) for 1 h at 25 °C. The cells were stained with mouse anti-Cy5 antibody (1:50, #ab52061, Abcam) for 16h at 25 °C, and incubated with Nanogold-IgG goat anti-mouse IgG (H + L) (1:200, #2002, Nanoprobes, Yaphank, NY, USA) for 2 h at 25 °C. Nanogold signals were enhanced with GoldEnhance™ EM (#2113, Nanoprobes) for 5 min at 25 °C. After fixation in 1% glutaraldehyde in 0.2 M HEPES, cells were post-fixed with 1% $OsO_4$ for 30 min at 4 °C and dehydrated through a graded ethanol series. The samples were incubated twice with propylene oxide for 15 min each, once with a 1:1 mixture of propylene oxide and epon for 4 h at 25 °C, once with a 1:3 mixture of propylene oxide and epon 16h at 25 °C, then embedded in pure epon for 3 days. Ultrathin sections (80 nm) were prepared with a ultramicrotome (UC7, Leica, Wetzlar, Germany) and stained with uranyl acetate and lead citrate. Ultrathin sections were observed by electron microscopy (JEM-1400, JEOL, Tokyo, Japan).

## Mutant Atxn1-KI mice

Mutant Atxn1-KI mouse (Sca1$^{154Q/2Q}$) is a generous gift from Prof. Huda Y. Zoghbi (Baylor College of Medicine, TX, USA) (Watase et al, 2002). The background mice (C57BL/6J) were used for breeding. The numbers of CAG repeat, which was checked by the following method, were changed after multiple times of crossing, and in this specific study, heterozygous KI mice with repeat numbers from 125 to 140 repeats were used. The mice were maintained at 22 °C, under suitable humidity (typically 50%), and with a 12-h dark/light cycle.

## Statistics and reproducibility

Statistical analyses for biological experiments were performed using Graphpad Prism 8. Biological data following a normal distribution are presented as the mean ± SEM, with Tukey's HSD test for multiple group comparisons or with Welch's t-test for two group comparisons. The distribution of observed data was depicted with box plots, with the data also plotted as dots. Box plots show the medians, quartiles, and whiskers, which represent data outside the 25th–75th percentile range. Data not following a normal distribution are examined by Wilcoxon's rank-sum test with post-hoc Bonferroni correction. To obtain each data, we performed biologically independent experiments. The number of samples was indicated in each figure and figure legends. No sample size calculation was performed, and the sample

size were similar to those reported in previous publications, PubMed ID 38594382 and 38017287. Simple randomization was performed to allocate samples and/or images to researchers before analysis. The selection of images from immunohistochemistry/immunocytochemisty and the actual experiments of IHC/ICC were done by different researchers. In vitro live-cell imaging were done by different researchers. Experiments for quantitative analyses were technically replicated until the necessary N was acquired. The information about group allocation or samples were opened to the data analyst or image acquisition researchers after finalizing results (make graphs etc).

## Ethics

This study was performed in strict accordance with the recommendations of the Guide for the Care and Use of Laboratory Animals of the Japanese Government and the National Institutes of Health. All animal experiments were performed in accordance with Animal Research: Reporting in vivo Experiments (ARRIVE) guidelines and were approved by the Committees on Gene Recombination Experiments and Animal Experiments of Tokyo Medical and Dental University (G2018-082C5 and A2023-113A).

# Data availability

All data generated or analyzed during this study are included in this article. Source data are provided with the paper.

The source data of this paper are collected in the following database record: biostudies:S-SCDT-10_1038-S44318-024-00192-4.

# Peer review information

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

## Acknowledgements

The authors thank Carl Zeiss Co., Ltd. and Yokogawa Electric Corporation for their generous support in conducting SRM. The authors also thank Dr. Xuemei Zhang for technical support (Tokyo Medical and Dental University). This work was supported by grants to HO, including a Grant-in-Aid for Scientific Research on Innovative Areas (Foundation of Synapse and Neurocircuit Pathology, 22110001/22110002) from the Ministry of Education, Culture, Sports, Science, and Technology of Japan (MEXT), a Grant of AMED-CREST from Japan Agency for Medical Research and Development (AMED) (JP24gm1910001) and a Grant-in-Aid for Scientific Research A (16H02655, 19H01042, 22H00464) from the Japanese Society for the Promotion of Science (JSPS).

## Author contributions

**Yuki Yoshioka**: Data curation; Formal analysis; Investigation. **Yong Huang**: Data curation; Formal analysis; Investigation. **Xiaocen Jin**: Data curation; Formal analysis; Investigation. **Kien Xuan Ngo**: Data curation; Investigation. **Tomohiro Kumaki**: Data curation; Investigation. **Meihua Jin**: Data curation; Investigation. **Saori Toyoda**: Data curation. **Sumire Takayama**: Data curation. **Maiko Inotsume**: Data curation. **Kyota Fujita**: Data curation; Formal analysis; Writing—original draft. **Hidenori Homma**: Software; Formal analysis; Investigation; Writing—original draft. **Toshio Ando**: Data curation; Investigation. **Hikari Tanaka**: Data curation; Formal analysis; Investigation; Methodology; Writing—original draft. **Hitoshi Okazawa**: Conceptualization; Supervision; Funding acquisition; Writing—original draft; Project administration; Writing—review and editing.

Source data underlying figure panels in this paper may have individual authorship assigned. Where available, figure panel/source data authorship is listed in the following database record: biostudies:S-SCDT-10_1038-S44318-024-00192-4.

## Disclosure and competing interests statement

The authors declare no competing interests.

# Expanded View Figures

**Figure EV1.  PQBP3/NOL7 puncti in the nucleoplasm and cytoplasm.**

(**A**) Diameters of small speckles of PQBP3/NOL7 were quantified in images obtained by SRM. Box plots show the median and 25–75th percentile, and whiskers represent data outside the 25–75th percentile range. (**B**) PQBP3/NOL7 is decreased in the nucleus and overall at high cell densities. Representative images of PQBP3/NOL7 and Hoechst 33342 containing at four different cell densities. (**C**) Quantitative analyses of PQBP3/NOL7 signal intensities per cell (left) and per nucleus (right) are shown in graphs. Box plots show the median and 25–75th percentile, and whiskers represent 1.5x inter-quartile range.  Source data are available online for this figure.

▶

**a**

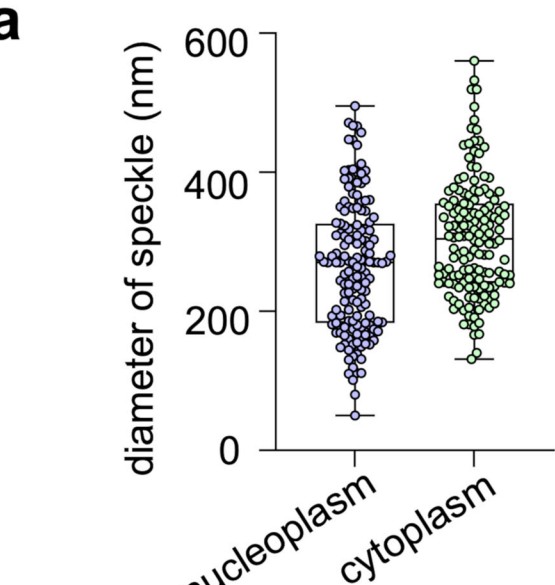

Average diameters
of small speckles

in nucleoplasm:
263.82 $\pm$ 7.65 nm (n=150)

In cytoplasm:
305.46 $\pm$ 6.84 nm (n=150)

**b**

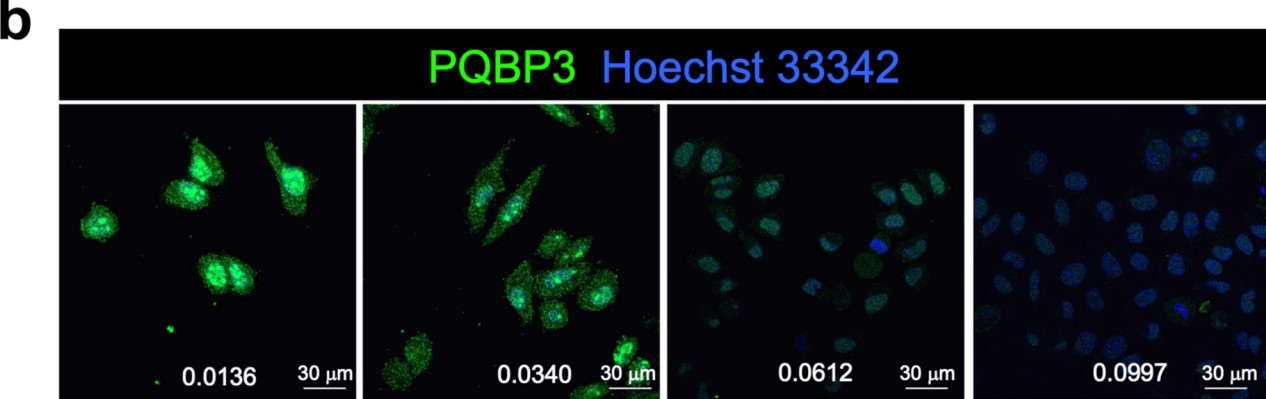

Cell number/100μm$^2$

**c**

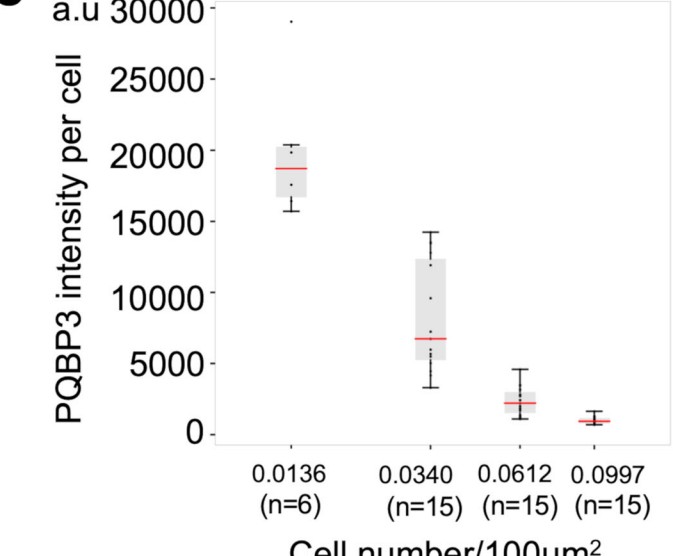

Kendall's rank correlation tau = - 0.830,
p = 1.113 × 10$^{-14}$

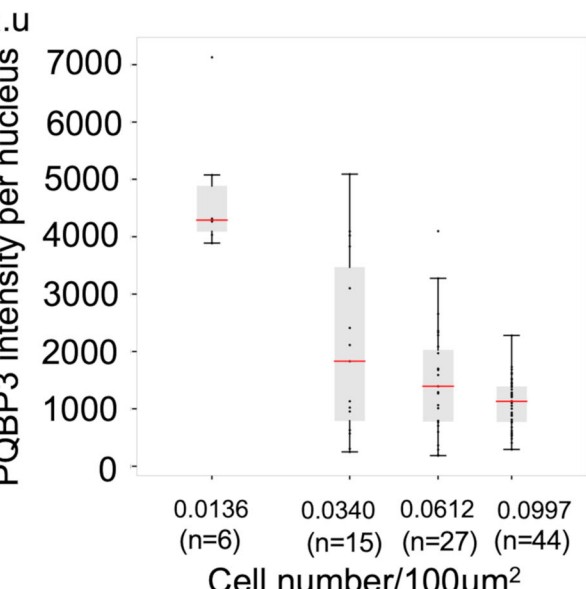

Kendall's rank correlation tau = - 0.316,
p = 8.768 × 10$^{-5}$

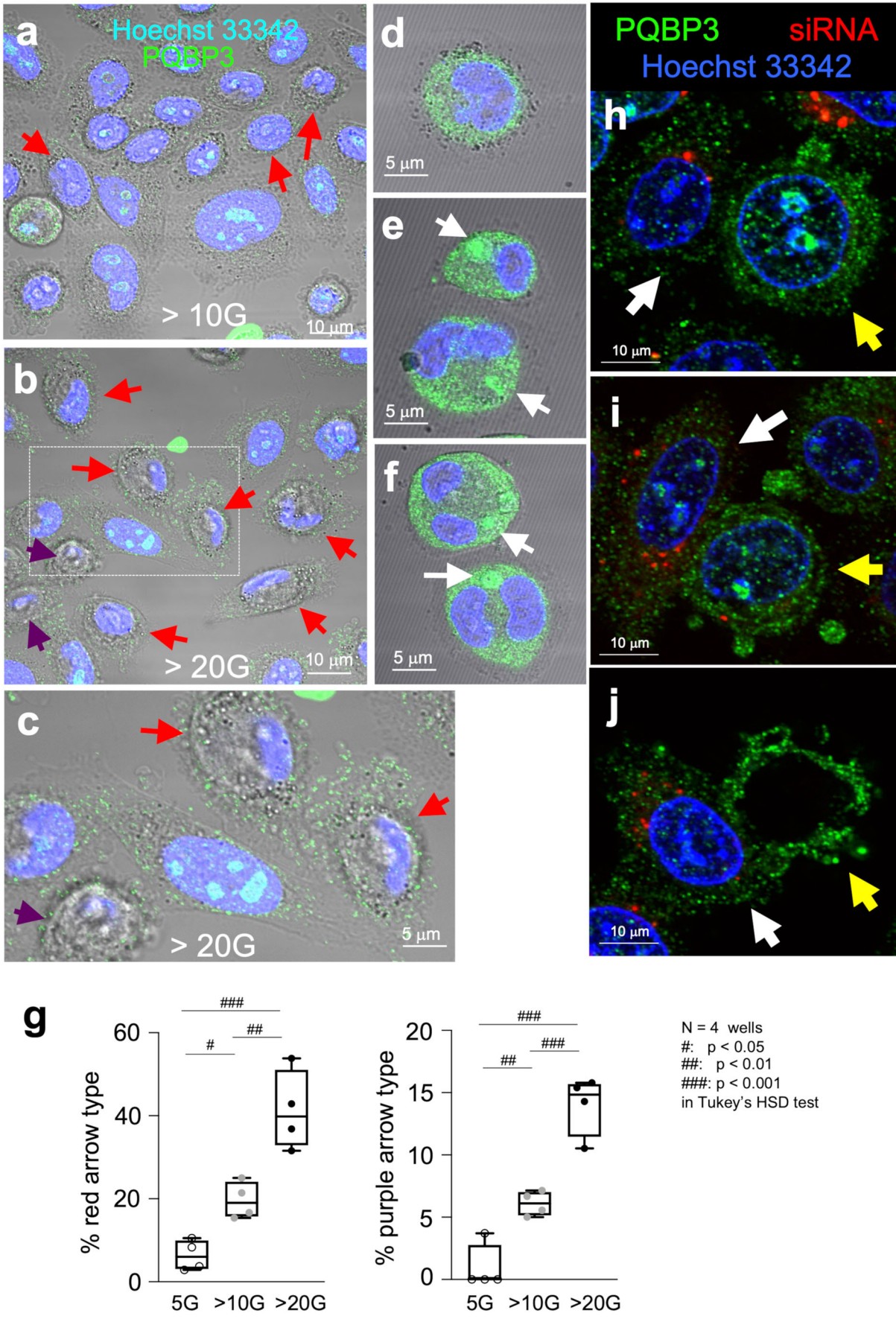

◀  **Figure EV2.   PQBP3/NOL7 in senescence.**

(**A**) Confocal microscopy of HeLa cells after ten passages (10G), which were penetrated with Tween20, immunostained with anti-PQBP1, and costained with Hoechst 33342. Red arrows indicate cells with dispersed nucleolar PQBP3/NOL7 staining. (**B**) Confocal microscopy images of HeLa cells after 20 passages (20G) stained as described above. Red arrows indicate cells with dispersed nucleolar PQBP3/NOL7 staining, and chromatin distribution (Hoechst 33342-stained area) shifted and deviated in the nucleus. Purple arrows indicate cells in which chromatin was nearly absent. (**C**) Enlarged image of the area indicated by dotted lines in (**B**). (**D–F**) Specific distributions of PQBP3/NOL7 during cell division. Foci of PQBP3/NOL7 localized to the centrosome (white arrow) were observed in addition to the diffuse cytoplasmic distribution. (**G**) Quantitative analyses of percentage of red arrow type or purple arrow type of cells in three different passage groups. Box plots show the median and 25–75th percentile, and whiskers represent data outside the 25–75th percentile range. In red arrow type of cells, statistical significance was found in comparison of <5G and >10G (#: $p = 0.0415$), <5G and >20G (###: $p < 0.0001$), and >10G and >20G (##: $p = 0.0027$). In purple arrow type of cells, statistical significance was found in comparison of <5G and >10G (##: $p = 0.0082$), <5G and >10G (###: $p < 0.0001$), and >10G and >20G (###: $p = 0.0005$). (**H–J**) Yellow arrows indicate cells exhibiting morphological changes of cell death, in which siPQBP3 signals (red) were absent or low, and PQBP3 signals (green) were relatively high. Contrastingly, siPQBP3-transfected cells with high red signals and low green signals did not exhibit such changes or apoptotic features (white arrows).  Source data are available online for this figure.

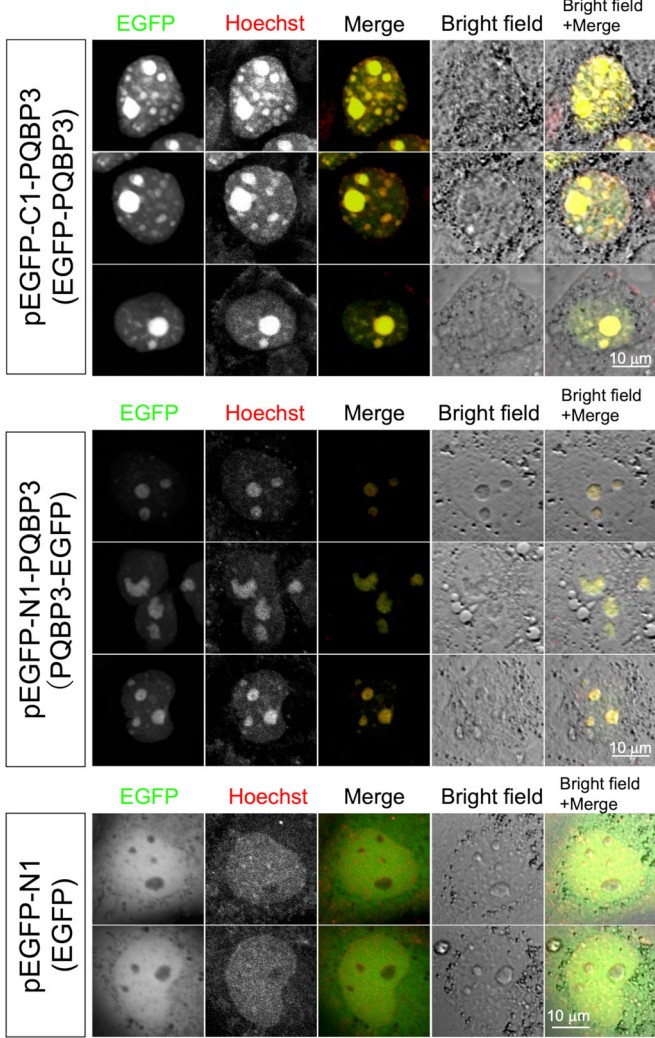

**Figure EV3.   Expression patterns of PQBP3/NOL7 fusion proteins.**

HeLa cells were transfected with pEGFP-C1-PQBP3, pEGFP-N1-PQBP3, or pEGFP-N1 plasmids to express EGFP-PQBP3, PQBP3-EGFP, or EGFP proteins, and following Hoechst 33342 staining without fixation, EGFP signals in live cells were observed with confocal microscopy. Similar expression patterns were observed in EGFP-PQBP3 and PQBP3-EGFP fusion proteins. EGFP protein alone did not exhibit the nucleolar pattern of the PQBP3 fusion proteins. Some cell images are redisplayed from Fig. 7B. Source data are available online for this figure.

**a**

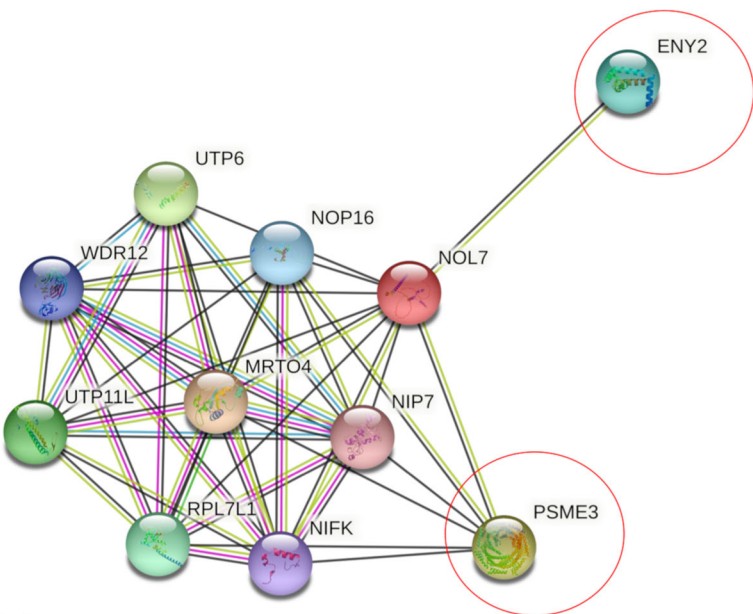

Information by String

**Proteasome activator complex subunit 3**; Subunit of the 11S REG-gamma (also called PA28-gamma) proteasome regulator, a doughnut-shaped homoheptamer which associates with the proteasome. 11S REG-gamma activates the trypsin-like catalytic subunit of the proteasome but inhibits the chymotrypsin-like and postglutamyl-preferring (PGPH) subunits. Facilitates the MDM2-p53/TP53 interaction which promotes ubiquitination- and MDM2-dependent proteasomal degradation of p53/TP53, limiting its accumulation and resulting in inhibited apoptosis after DNA damage. May also be involved in cell cycle regul [...]
Identifier: ENSP00000293362, **PSME3**
Organism: Homo sapiens

**Transcription and mRNA export factor ENY2**; Involved in mRNA export coupled transcription activation by association with both the TREX-2 and the SAGA complexes. The transcription regulatory histone acetylation (HAT) complex SAGA is a multiprotein complex that activates transcription by remodeling chromatin and mediating histone acetylation and deubiquitination. Within the SAGA complex, participates in a subcomplex that specifically deubiquitinates both histones H2A and H2B. The SAGA complex is recruited to specific gene promoters by activators such as MYC, where it is required for trans [...]
Identifier: ENSP00000429986, **ENY2**
Organism: Homo sapiens

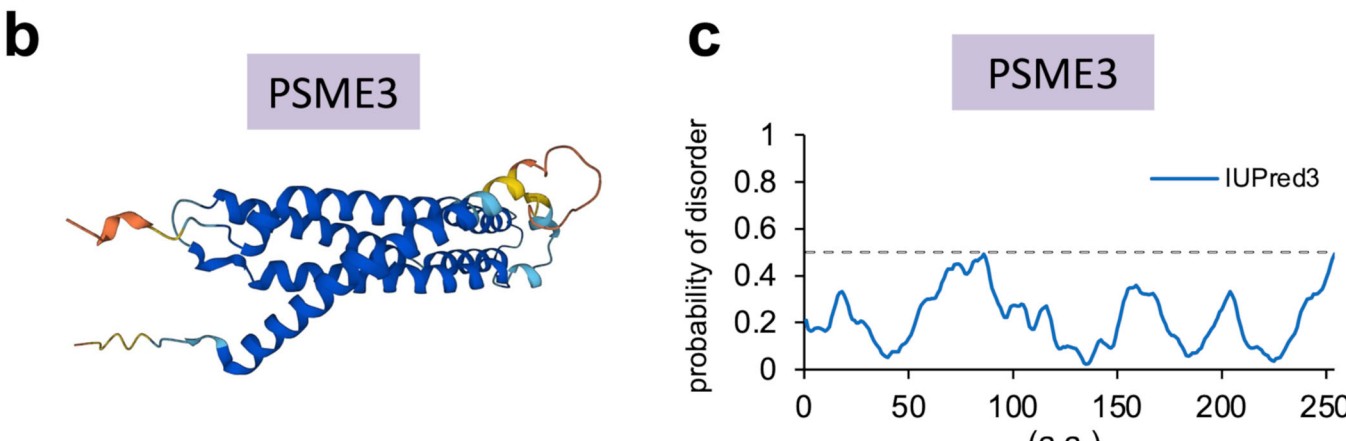

**b** PSME3

https://alphafold.ebi.ac.uk/entry/Q967U1

**c** PSME3

**IUPred2A** (https://iupred2a.elte.hu/)

**Figure EV4.   PSME3 as a candidate proteins interacting with PQBP3/NOL7.**

(**A**) String (version 11.5) (https://string-db.org/) was used to predict interacting proteins with PQBP3/NOL7, and their descriptions in String are shown. (**B**) PSME3 protein structure predicted by alphafold. (**C**) PSME3 IDP prediction by IUPred2A.

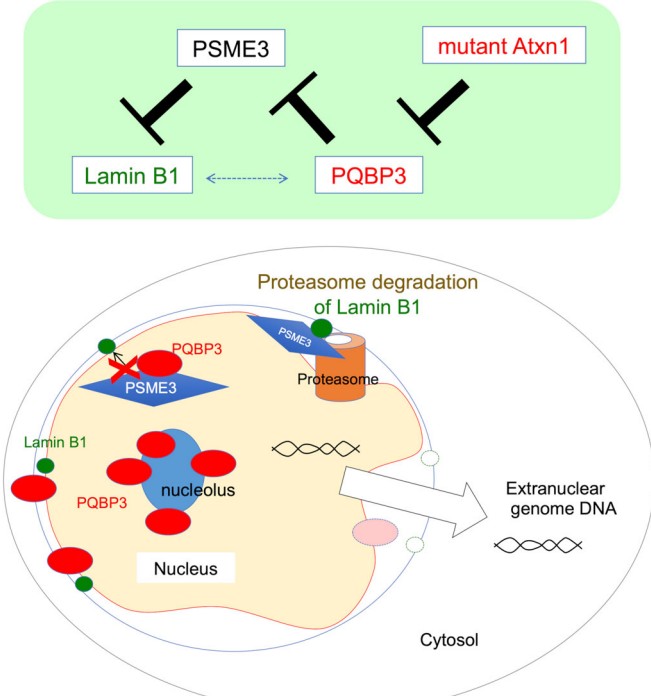

**Figure EV5.　Hypothesized mechanism for nuclear membrane instability mediated by PQBP3/NOL7 and PSME3.**

Hypothesized mechanism of nuclear membrane instability under senescence, as suggested by the results of the present study. The upper panel illustrates the interaction and suppression relationships between PQBP3/NOL7, PSME3, Lamin B1, and mutant Atxn1. Under physiological conditions, PQBP3 complexes with PSME3 to suppress its protein degradation activity. In senescence, PQBP3 is decreased and not supplied sufficiently to the nuclear membrane for inhibition of PSME3-mediated proteasomal degradation of Lamin B1. In the case Lamin B1 is degraded, the nuclear membrane is instabilized, allowing release of genomic DNA from the nucleus to the cytosol.

