## [Peer Review File · The EMBO Journal]

PQBP3 prevents senescence by suppressing PSME3-mediated proteasomal Lamin B1 degradation

Yuki Yoshioka, Yong Huang, Xiaocen Jin, Kien Ngo, Tomohiro Kumaki, Meihua Jin, Saori Toyoda, Sumire Takayama, Maiko Inotsume, Kyota Fujita, Hidenori Homma, Toshio Ando, Hikari Tanaka, and Hitoshi Okazawa

Corresponding author: Hitoshi Okazawa (okazawa.npat@mri.tmd.ac.jp)

Review Timeline:

Submission Date:	22nd Jun 23
Editorial Decision:	27th Jul 23
Revision Received:	23rd May 24
Editorial Decision:	4th Jul 24
Revision Received:	12th Jul 24
Accepted:	22nd Jul 24

Editor: Daniel Klimmeck

Transaction Report:

Dear Dr Okazawa,

Thank you for submitting your manuscript for consideration by the EMBO Journal. It has now been seen by three referees whose comments are shown below.

Should you be able to address these criticisms - detailing important limitations regarding the choice of appropriate cellular models, robustness of the biochemical data presented and completeness of the senescence characterisation - in full, we could consider a revised manuscript. I should remind you that it is EMBO Journal policy to allow a single round of revision only and that, therefore, acceptance or rejection of the manuscript will depend on the completeness of your responses in this revised version. I do realize that addressing all the referees' criticisms will require a lot of additional time and effort and be technically challenging.

If you decide to thoroughly revise the manuscript for the EMBO Journal, please include a detailed point-by-point response to the referees' comments. Please bear in mind that this will form part of the Review Process File, and will therefore be available online to the community. For more details on our Transparent Editorial Process, please visit our website: <https://www.embo.org/embo-press>

Thank you for the opportunity to consider your work for publication. I look forward to your revision.

Kind regards,

Daniel Klimmeck

Daniel Klimmeck, PhD
Senior Editor
The EMBO Journal

Instruction for the preparation of your revised manuscript:

2) individual production quality figure files as .eps, .tif, .jpg (one file per figure).

3) a .docx formatted letter INCLUDING the reviewers' reports and your detailed point-by-point response to their comments. As part of the EMBO Press transparent editorial process, the point-by-point response is part of the Review Process File (RPF), which will be published alongside your paper.

4) a complete author checklist, which you can download from our author guidelines ([https://wol-prod-cdn.literatumonline.com/pb-assets/embo-site/Author Checklist%20-%20EMBO%20J-1561436015657.xlsx](https://wol-prod-cdn.literatumonline.com/pb-assets/embo-site/Author%20Checklist%20-%20EMBO%20J-1561436015657.xlsx)). Please insert information in the checklist that is also reflected in the manuscript. The completed author checklist will also be part of the RPF.

6) It is mandatory to include a 'Data Availability' section after the Materials and Methods. Before submitting your revision, primary datasets produced in this study need to be deposited in an appropriate public database, and the accession numbers and database listed under 'Data Availability'. Please remember to provide a reviewer password if the datasets are not yet public (see <https://www.embopress.org/page/journal/14602075/authorguide#datadeposition>).

7) Our journal encourages inclusion of *data citations in the reference list* to directly cite datasets that were re-used and obtained from public databases. Data citations in the article text are distinct from normal bibliographical citations and should directly link to the database records from which the data can be accessed. In the main text, data citations are formatted as follows: "Data ref: Smith et al, 2001" or "Data ref: NCBI Sequence Read Archive PRJNA342805, 2017". In the Reference list, data citations must be labeled with "[DATASET]". A data reference must provide the database name, accession number/identifiers and a resolvable link to the landing page from which the data can be accessed at the end of the reference. Further instructions are available at .

8) At EMBO Press we ask authors to provide source data for the main and EV figures. Our source data coordinator will contact you to discuss which figure panels we would need source data for and will also provide you with helpful tips on how to upload and organize the files.

Numerical data can be provided as individual .xls or .csv files (including a tab describing the data). For 'blots' or microscopy, uncropped images should be submitted (using a zip archive or a single pdf per main figure if multiple images need to be supplied for one panel). Additional information on source data and instruction on how to label the files are available at .

9) We replaced Supplementary Information with Expanded View (EV) Figures and Tables that are collapsible/expandable online (see examples in <https://www.embopress.org/doi/10.15252/emboj.201695874>). A maximum of 5 EV Figures can be typeset. EV Figures should be cited as 'Figure EV1, Figure EV2' etc. in the text and their respective legends should be included in the main text after the legends of regular figures.

10) When assembling figures, please refer to our figure preparation guideline in order to ensure proper formatting and readability in print as well as on screen:
<http://bit.ly/EMBOPressFigurePreparationGuideline>

11) For data quantification: please specify the name of the statistical test used to generate error bars and P values, the number (n) of independent experiments (specify technical or biological replicates) underlying each data point and the test used to calculate p-values in each figure legend. The figure legends should contain a basic description of n, P and the test applied. Graphs must include a description of the bars and the error bars (s.d., s.e.m.).

The revision must be submitted online within 90 days; please click on the link below to submit the revision online before 25th Oct 2023.

Referee #1:

The authors conclude "PQBP3/NOL7 suppresses proteasome activator complex subunit 3 (PSME3).....to prevent Lamin B1 degradation. Deficiency of PQBP3/NOL7 in the nucleolus causes nuclear membrane instability and release of genomic DNA from the nucleus to the cytosol." There are many deficiencies in the data that preclude this conclusion.

1. The origin of the cytoplasmic fragments in siPQB3 cells is not clear. Do they originate as micronuclei in failed mitosis or rupturing of chromatin through the nuclear envelope?
2. The Western blots in Figure 7 are key to the proposed mechanism, but these are generally poor quality.
3. All the Western blot results with PQBP3 are based on an ectopic expression EGFP fusion protein. There are no data from endogenous PQBP3.
4. The identity of the ubiquitinated lamin B1 band is not rigorously confirmed.
5. There is insufficient evidence to indicate that PSME3 is involved in lamin B1 degradation.

Referee #2:

Senescence of non-dividing cells such as neurons could have implications for tissue homeostasis and disease. However, how senescence of neurons is triggered remains largely unknown. This is a very interesting study that provides insights into the molecular mechanisms that trigger neuronal senescence and their links with polyQ-related neurodegenerative diseases. First, the authors discovered that polyglutamine-binding protein 3 (PQBP3)-a protein that binds polyQ repeats- regulates senescence through its role as a gatekeeper of DNA leakage to the cytosol. At the mechanistic level, I find particularly interesting the links with PSME3. Here, the authors show that PQBP3 suppresses PSME3 to prevent lamin B1 degradation. The role of PSME3 in regulating Lamin B1 levels and senescence is very relevant to understand the cellular role of PSME3. Interestingly, the authors' findings indicate that deficiency of PQBP3 leads to nuclear membrane instability and subsequent release of genomic DNA to the cytosol. In addition, the authors also provide data on the physiological consequences of alterations in this regulatory mechanism. To this end, the authors tested multiple polyQ-containing proteins. Notably, they found that ataxin-1 partially sequesters PQBP3 into aggregates, reducing the levels of PQBP3 in the nucleolus. Finally, their results suggest that a SCA1 mouse model expressing ataxin-1 with 154 polyQ repeats have decreased levels of PQBP3 in the nucleus and a senescence phenotype in Purkinje cells. Together, this study is relevant to understand senescence of neurons and its implications for neurodegenerative disorders. Nevertheless, I have several comments that I hope the authors will be able to address:

Major points:

- The focus of the paper is cell senescence, with a particular interest in senescence of non-dividing cells. Most of the main conclusions of the paper are based on experiments in HeLa cells and inhibition of their cell proliferation. They first reduced cell proliferation through cell confluency. As the authors indicate confluency suppresses cell proliferation but does not induce senescence. Along these lines, the authors use a whole paragraph to explain the role of mTOR activity in activation of senescence, but they do not assess experimentally whether activation of mTOR activity could induce senescence in their model and lead to changes in PQBP3 similar to what they observed upon inhibition of proliferation. Such experiments would be very relevant to strengthen the conclusions of the manuscript. Importantly, they also examined HeLa cells after several passages to inhibit cell proliferation, which could be a more suitable model to study senescence. Nevertheless, to support the conclusions of the paper and the links with senescence, the authors could also induce senescence through more direct means. For instance, mTOR as indicated above or etoposide (although perhaps it would be difficult to interpret the data since senescence-induction drugs can have strong cell death effects in immortalized cell lines).
- Besides arrested HeLa cells and given the focus of the paper on senescence of non-dividing cells, it is critical that the authors assess whether the most important conclusions (Figs. 2-7) also occur in cultures of non-dividing, terminally differentiated cells such as neurons or muscle cells. Could it be possible to use established protocols to induce senescence in cultures of terminally differentiated cells in vitro (e.g. by keeping the cultures for extended periods of time or treatment with drugs such as etoposide)? For instance, data from primary cultures, differentiated cells-derived from iPSCs would really strengthen the conclusions of the paper. If this is not possible, the authors could consider to differentiate muscle or neural cell lines such as C2C12 or SH-SY5Y.
- In Figure 3/4, the authors need to support the conclusions regarding the changes in the levels of nucleolar and total levels PQBP3 with western blots of total cell lysates and, if possible, of nuclear and cytosol fractions.
- In the results section for Figure 4, the authors mention: "A portion of the cells exhibited cell death-like changes with multiple protrusions of cytosol (Figure 4a, purple arrow)". However, I cannot see a purple arrow in Figure 4A (do they mean Figure 4b?). Related with this point, I cannot understand the following conclusion: "These cells lost nucleolar PQBP3/NOL7 staining but exhibit increased cytosolic PQBP3/NOL7 speckles (Figure 4a, yellow, red, purple arrows), suggesting that diminishment of nucleolar PQBP3/NOL7 was due to decreased synthesis or increased degradation of PQBP3/NOL7 rather than shifting of PQBP3/NOL7 from the nucleus to the cytoplasm". First, I interpret that the authors want to mention Figure 4b instead of 4a? Second, is not their conclusion exactly the opposite of what they want to say and can be interpreted from the data? If there is a

decrease in nucleolus PQBP3 while its cytosolic levels increase, these data would suggest changes in subcellular localization rather than changes in translation or degradation of PQBP3. Please revise the text for this part. Most importantly, western blots of total cells and cellular fractions (nucleus and cytosol) are needed to support the conclusions.

- The authors mention: "A previous study suggested that PQBP3/NOL7 knockdown did not significantly affect nucleolus or cell structure in H1299 human lung carcinoma cells(Kinor & Shav-Tal, 2011), while the experiment was rather artificial as they overexpressed PQBP3/NOL7-YFP and subsequently knocked down the protein by siRNA(Kinor & Shav-Tal, 2011). The insufficiency prompted us to re-examine the effect of PQBP3/NOL7 knockdown on nucleolus. In addition, we investigated how knockdown of endogenous PQBP3/NOL7 affected cell morphology and cytoplasmic DNA, and whether the phenotype was relevant to senescence (Figure 5)". I recommend the authors to avoid using terms such as artificial or insufficiency.
- To obtain conclusions from experiments using siRNAs (Figure 5), at least 2 independent siRNA constructs should be tested to discard off-target effects.
- In Figure 5, the authors focus on nuclear shape and cytosolic DNA. It is critical for the conclusions of the paper that the authors assess by B-gal staining whether knockdown of PQBP3 indeed induces senescence in HeLa cells and terminally differentiated cells. Likewise, B-gal staining should be tested to assess whether PQBP3 overexpression rescues senescence.
- The authors should address many parts of the text as sometimes it's difficult to follow what they say (repeated sentences, typos and cut sentences through the text). For instance: "We postulated that PQBP3/NOL7 could function as a positive or negative regulator of PSME3, a proteasomal activator, affecting degradation of senescence-related proteins such as Lamin B1, which plays critical roles in maintaining the nuclear shape(Shimi et al, 2008) the hydrogen peroxide protocol, and investigated the relationship between PQBP3/NOL7, PSME3, and LaminB1" or "Prior studies have demonstrated changes to the ubiquitin-proteasome system during senescence(Deschênes-Simard et al, 2014; Fukuura et al, 2019; Marfella et al, 2008; Ullah et al, 2020), and total protein ubiquitination is decreased during aging, and total protein ubiquitination is decreased during aging in lower animals such as *C. elegans*(Koyuncu et al, 2021), consistent with our findings", or "functional site of PQBP5/NOL10(Jin et al, 2023). To determine if similar sequestration of PQBP3/NOL3 to polyQ. To determine if similar sequestration of PQBP3/NOL3 to polyQ inclusion bodies occurred, we examined intracellular localization patterns of polyQ disease proteins and PQBP3/NOL3 in their coexpression (Figure 8a, b)".
- using immunocytochemistry and immunoprecipitation (Figure 7a). using immunocytochemistry and immunoprecipitation (Figure 7a).
- PSME3 appears to degrade proteins in a ubiquitin-independent manner (for review please see Mao et al Cell. Mol. Life Sci. 65, 3971-3980 (2008)). Can the authors discuss how this could fit with their data in Figure 7?

Minor points:

- The Introduction is very comprehensive, but it contains some information that it is not so relevant to understand the present study. The authors might think about shorten it to make it more straightforward to the topic of the study.
- Along these lines, I don't think the authors need to spend almost an entire paragraph to explain the differences between IUPred2A and RONN (a software that it's no longer available and the authors couldn't use). I would just focus on the analysis and conclusions from IUPred2A and remove all the discussion about RONN.
- After talking about the analysis using IUPred2A, they directly start the following paragraph with: "Glutathione S-transferase (GST) protein was cleaved from the GST-PQBP3 fusion protein (GST-PQBP3), and the cleaved PQBP3/NOL7 protein was subjected to high-speed atomic force microscopy (HS-AFM) for direct observation of dynamic protein structure changes". A better connection between paragraphs or context would help the readability of the paper. (for instance, starting the sentence with Glutathione S-transferase (GST) protein might be not so straightforward)
- Recent studies have showed a role of PSME-activated proteasomes in the degradation of misfolded and disease-related proteins which are prone to misfold and aggregate including polyQ-expanded proteins (Frayssinhes et al, Cell. Mol. Life Sci. 2022, Lee et al, Nature Aging 2023). Here the authors discovered a link between PSME3, lamin B1 degradation and polyQ diseases. Given the role of PSME3 in degrading polyQ-expanded proteins, it would be interesting if the authors discuss these findings in the context of their results and the potential links with senescence.

Referee #3:

Jin et al. present an interesting study investigating the relationship between the nucleolus and senescence, specifically pertaining the role of PQBP3/NOL7 in protecting cells against senescence. The authors propose that loss of NOL7 function affects LaminB1, leading to cytoplasmic DNA accumulation which drives the senescence phenotype. They further suggest NOL7 is sequestered in ATXN1 aggregates in models of spinocerebellar ataxia, and that therefore Purkinje cells may suffer a similar fate. Overall, the experimental work is rigorous-though should be more quantitative in nature-and the manuscript is well-written. I do have some issues with the work in its current form, which I address in detail below. If the authors are able to address these comments, I would be supportive for publication of the work in EMBO journal.

Major comments:

1. The study falls a bit short on the discussion and assessment of senescence. As the authors are aware, there are many different paradigms and discrepancies in the field. The choice of system should be discussed in more detail and characterized more. Neuronal senescence occurs not only due to normal aging, but also due to damage and other stress stimuli. Therefore, it is important for the authors to define and discuss their definition of senescence. For example, the authors assess contact inhibition and assume cellular quiescence/senescence, but there are better ways to define a quiescent or senescent cell on a single-cell level. For example, the nuclear to cytoplasmic ratio of DHB is often used as an indication of G0-as DHB is nuclear during quiescence. Since the study aims to understand neuronal senescence in disease it is important to study if the same molecular interactions in damage-induced senescence and not only replicative senescence. Additionally, the idea that mTOR activity can define senescence versus quiescence is not specific enough. mTOR is implicated in many pathways and the fluctuation of its activity cannot define the state. Although high mTOR activity may be true for quiescent cells and mTOR may increase in senescence, other markers in tandem within the same cell would be required to confidently define senescence versus quiescence.
2. Figure 1 is distracting and completely irrelevant for this study. Whether NOL7 is a disordered protein or not contributes nothing to the rest of the work. The authors simply show some disorder prediction plots. The AFM data also does not add, but raises A LOT of questions. We see one supposedly representative example of a tadpole structure. What this means for NOL7's role in senescence is unclear. Whether this structure is physiological is unclear. The authors have the choice: (1) they omit the entirety of Figure 1 from the work as it adds no value, is distracting and highly preliminary or (2) they perform an in-depth structural characterization of NOL7 and link its disorder to its actual function. The authors have a great story here, but this figure detracts from the entirety of the work. I would strongly advise the authors to cut this data and pursue such studies in follow-up work. IDPs have been studied for decades. Showing that a protein is an IDP without studying the functional consequences of that is not something that is of interest to the field in or benefits this study. I also disagree with having the very preliminary AFM data piggyback on the rest of the work. There are too many questions here for it to be published in its current form.
3. Figure 4 is hard to interpret because of the DIC overlay. In general, the authors should also perform actual quantifications of the phenotypes they observe, rather than simply relying on representative images. This is true for several figures in the manuscript. For example, here the authors should quantify nucleolar and cytoplasmic NOL7. Also, beta galactosidase should be quantified and correlated with nucleolar NOL7.
4. The authors discuss SAHFs as being an indispensable phenotype for senescent cells, but they do not confirm that a knockdown of NOL7 induces this phenotype. In prior figures, they only measured senescent markers with replicative senescence, but here we are looking at the supposed senescence induced by NOL7 knock-down.
5. Figure 7: The authors should validate that their hydrogen peroxide protocol induces senescence by evaluating established markers.
6. Figure 8: This figure is somewhat puzzling. In cells we do see NOL7 recruitment to ATXN1 assemblies, but this is not seen (or not clearly shown) in ATXN1 mice. The authors discuss in length that the sequestration of NOL7 in polyQ aggregates is a novel toxic mechanism in the conditions, yet currently they do not provide any evidence for this sequestration. Overall, the images here should also be quantified, rather than just showing a single representative example.
7. Discussion: "Elucidating the full mechanism of the pathology like "digital twin" at the molecule or cell level could be accomplished using mathematical simulations based on integration of comprehensive data from multiomics and the LLPS-based interactome, which could be advanced from a prototype based on the proteome(Jin et al, 2021a)." This sentence is non-sensical and pretty much just a bunch of buzzwords thrown together, boiling down on the vague promise that one day computer simulations will recreate reliable and useful models of cells that allow us to supposedly solve disease. I have literally no clue what the authors mean with this statement or how it relates to any of the work. What is an "LLPS-based interactome" even? As someone who has studied this phenomenon for nearly a decade I have no idea what the authors are referring to. Please remove, it is off-putting and has zero relevance to the data and the rest of the manuscript.

Referee #1:

The authors conclude "PQBP3/NOL7 suppresses proteasome activator complex subunit 3 (PSME3).....to prevent Lamin B1 degradation. Deficiency of PQBP3/NOL7 in the nucleolus causes nuclear membrane instability and release of genomic DNA from the nucleus to the cytosol." There are many deficiencies in the data that preclude this conclusion.

>>> We thank very much Referee 1 for the critical comments. We could strengthen the basis for our conclusion via responding to the comments.

1. The origin of the cytoplasmic fragments in siPQB3 cells is not clear. Do they originate as micronuclei in failed mitosis or rupturing of chromatin through the nuclear envelope?

>>> Thank you very much for this critical comment to point out our logic contradiction. As the reviewer #1 mentioned, our investigation started from the finding of increased "what we called micronuclei" in siPQBP3-transfected cells (Figure 3), but later we claimed that PQBP3 reduction is associated DNA leakage from nucleus to cytoplasm (Figure 6, 7). However, during the progress of our story, we showed both phenotypes were increased by si-PQBP3 (Figure 5). Figure 5a, c(right graph) and d show increased leakage, while Figure 5b and c show increased " what we called micronuclei".

The reviewer's critical comment concerned such a confusing interpretation, and has suggested that "what we called micronuclei" looked a protrusion of the nucleus rather than a small micronucleus separated from the original nucleus. Therefore, we performed

electron microscopy and observed whether the nuclear membrane is disrupted or not, and how the chromatin looks like in such cells with a protrusion.

The results revealed that they are protrusion of nuclear membrane due to the nuclear membrane instability rather than separated micronuclei (new Figure 2e).

2. The Western blots in Figure 7 are key to the proposed mechanism, but these are generally poor quality.

>>> Thank you very much. The comment is exactly true. We repeated the WB experiments and improved the data in Figure 7 as possible as we could.

3. All the Western blot results with PQBP3 are based on an ectopic expression EGFP fusion protein. There are no data from endogenous PQBP3.

>>> We performed IP with endogenous PQBP3 (new Figure 8e).

4. The identity of the ubiquitinated lamin B1 band is not rigorously confirmed.

>>> We performed IP and confirmed that the Lamin B1 band is ubiquitinated. Interestingly, we found that the Lamin was also SUMOylated simultaneously.

5. There is insufficient evidence to indicate that PSME3 is involved in lamin B1 degradation.

>>> We performed knockdown and overexpression of PSME3 and observed the quantitative effects on Lamin B1. The result supported that PSME3 is involved in Lamin B1 degradation. We also revealed that SUMOylated Lamin B1 was the target of PSME3-activated proteasomal degradation.

Referee #2:

Senescence of non-dividing cells such as neurons could have implications for tissue homeostasis and disease. However, how senescence of neurons is triggered remains largely unknown. This is a very interesting study that provides insights into the molecular mechanisms that trigger neuronal senescence and their links with polyQ-related neurodegenerative diseases. First, the authors discovered that polyglutamine-binding protein 3 (PQBP3)-a protein that binds polyQ repeats- regulates senescence through its role as a gatekeeper of DNA leakage to the cytosol. At the mechanistic level, I find particularly interesting the links with PSME3. Here, the authors show that PQBP3 suppresses PSME3 to prevent lamin B1 degradation. The role of PSME3 in regulating Lamin B1 levels and senescence is very relevant to understand the cellular role of PSME3. Interestingly, the authors' findings indicate that deficiency of PQBP3 leads to nuclear membrane instability and subsequent release of genomic DNA to the cytosol. In addition, the authors also provide data on the physiological consequences of alterations in this regulatory mechanism. To this end, the authors tested multiple polyQ-containing proteins. Notably, they found that ataxin-1 partially sequesters PQBP3 into aggregates, reducing the levels of PQBP3 in the nucleolus. Finally, their results suggest that a SCA1 mouse model expressing ataxin-1 with 154 polyQ repeats has decreased levels of PQBP3 in the nucleus and a senescence phenotype in Purkinje cells. Together, this study is relevant to understand senescence of neurons and its implications for neurodegenerative disorders. Nevertheless, I have several comments that I hope the authors will be able to address:

>>> We thank very much kind evaluation of our manuscript and critical comments.

Major points:

- The focus of the paper is cell senescence, with a particular interest in senescence of non-dividing cells. Most of the main conclusions of the paper are based on experiments in HeLa cells and inhibition of their cell proliferation. They first reduced cell proliferation through cell confluency. As the authors indicate confluency suppresses cell proliferation but does not induce senescence. Along these lines, the authors use a whole paragraph to explain the role of mTOR activity in activation of senescence, but they do not assess experimentally whether activation of mTOR activity could induce senescence in their model and lead to changes in PQBP3 similar to what they observed upon inhibition of proliferation. Such experiments would be very relevant to strengthen the conclusions of the manuscript. Importantly, they also examined HeLa cells after several passages to inhibit cell proliferation, which could be a more suitable model to study senescence. Nevertheless, to support the conclusions of the paper and the links with senescence, the authors could also induce senescence through more direct means. For instance, mTOR as indicated above or etoposide (although perhaps it would be difficult to interpret the data since senescence-induction drugs can have strong cell death effects in immortalized cell lines).

>>> We performed experiments with a mTOR activator MHY1485 (new Figure 4). Treatment with MHY1485 decreased PQBP3 in WB and increased reactivity to beta-Gal in immunocytochemistry of HeLa cells.

- Besides arrested HeLa cells and given the focus of the paper on senescence of non-dividing cells, it is critical that the authors assess

whether the most important conclusions (Figs. 2-7) also occur in cultures of non-dividing, terminally differentiated cells such as neurons or muscle cells. Could it be possible to use established protocols to induce senescence in cultures of terminally differentiated cells in vitro (e.g. by keeping the cultures for extended periods of time or treatment with drugs such as etoposide)? For instance, data from primary cultures, differentiated cells-derived from iPSCs would really strengthen the conclusions of the paper. If this is not possible, the authors could consider to differentiate muscle or neural cell lines such as C2C12 or SH-SY5Y.

>>> Following the advice of the reviewer #2, we performed experiments with human iPSC-derived neurons. We confirmed by super resolution microscopy the similar nucleolar distribution patterns of PQBP3 in human iPSC-derived neurons (new Figure 1), and confirmed that hydrogen peroxide induced cytoplasmic shift of PQBP3 as well as senescence in human iPSC-derived neurons (new Figure 6), and revealed that PQBP3-siRNA transfection induced senescence in human iPSC-derived neurons (new Figure 6).

- In Figure 3/4, the authors need to support the conclusions regarding the changes in the levels of nucleolar and total levels PQBP3 with western blots of total cell lysates and, if possible, of nuclear and cytosol fractions.

>>> We followed the advice and performed WBs with nuclear and cytosolic fractions, and showed the results in new Figure 2c.

- In the results section for Figure 4, the authors mention: "A portion of the cells exhibited cell death-like changes with multiple protrusions of cytosol (Figure 4a, purple arrow)". However, I cannot see a purple arrow in Figure 4A (do they mean Figure 4b?).

>>> As the reviewer mentioned, it was our error and we meant Figure 4b and c (previous Fig numbers).

Related with this point, I cannot understand the following conclusion: "These cells lost nucleolar PQBP3/NOL7 staining but exhibit increased cytosolic PQBP3/NOL7 speckles (Figure 4a, yellow, red, purple arrows), suggesting that diminishment of nucleolar PQBP3/NOL7 was due to decreased synthesis or increased degradation of PQBP3/NOL7 rather than shifting of PQBP3/NOL7 from the nucleus to the cytoplasm". First, I interpret that the authors want to mention Figure 4b instead of 4a?

Second, is not their conclusion exactly the opposite of what they want to say and can be interpreted from the data? If there is a decrease in nucleolus PQBP3 while its cytosolic levels increase, these data would suggest changes in subcellular localization rather than changes in translation or degradation of PQBP3. Please revise the text for this part. Most importantly, western blots of total cells and cellular fractions (nucleus and cytosol) are needed to support the conclusions.

>>> We agree that this small conclusion was too careless and that the data here might have suggested the opposite. We confirmed it by western blots with total cell extract, nuclear fraction, and cytosolic fraction, and showed the data (new Figure 2c). As expected by the reviewer, cytoplasmic shift is a major contributor to morphological changes.

We moved the old Figure 4 to new Figure EV2.

- The authors mention: "A previous study suggested that PQBP3/NOL7 knockdown did not significantly affect nucleolus or cell

structure in H1299 human lung carcinoma cells(Kinor & Shav-Tal, 2011), while the experiment was rather artificial as they overexpressed PQBP3/NOL7-YFP and subsequently knocked down the protein by siRNA(Kinor & Shav-Tal, 2011). The insufficiency prompted us to re-examine the effect of PQBP3/NOL7 knockdown on nucleolus. In addition, we investigated how knockdown of endogenous PQBP3/NOL7 affected cell morphology and cytoplasmic DNA, and whether the phenotype was relevant to senescence (Figure 5)". I recommend the authors to avoid using terms such as artificial or insufficiency.

>>> We followed the advice, and deleted such words.

- To obtain conclusions from experiments using siRNAs (Figure 5), at least 2 independent siRNA constructs should be tested to discard off-target effects.

>>> We used added another siRNA and showed that the two siRNAs possess the similar effect in new Figure 5.

- In Figure 5, the authors focus on nuclear shape and cytosolic DNA. It is critical for the conclusions of the paper that the authors assess by B-gal staining whether knockdown of PQBP3 indeed induces senescence in HeLa cells and terminally differentiated cells. Likewise, B-gal staining should be tested to assess whether PQBP3 overexpression rescues senescence.

>>> As the reviewer #2 mentioned, our claim will become more solid if PQBP3-KD also induces a senescence marker b-Gal. We added beta-Gal staining of siPQBP3-transfected HeLa cells and of siPQBP3-transfected iPSC-derived neurons in new Figure 6.

Regarding senescence of terminally differentiated cells, there would be some discussion, as the reviewer #2 may know. B-Gal signals exist in normal neurons from the beginning (for instance, see Wang D-X et al, Nature Commun 2023; Piechota M et al, Oncotarget 2016). Therefore, so called “neuronal senescence” is discussed based on the quantitative change of signal intensities of b-Gal.

- The authors should address many parts of the text as sometimes it's difficult to follow what they say (repeated sentences, typos and cut sentences through the text). For instance: "We postulated that PQBP3/NOL7 could function as a positive or negative regulator of PSME3, a proteasomal activator, affecting degradation of senescence-related proteins such as Lamin B1, which plays critical roles in maintaining the nuclear shape(Shimi et al, 2008) the hydrogen peroxide protocol, and investigated the relationship between PQBP3/NOL7, PSME3, and LaminB1" or "Prior studies have demonstrated changes to the ubiquitin-proteasome system during senescence(Deschênes-Simard et al, 2014; Fukuura et al, 2019; Marfella et al, 2008; Ullah et al, 2020), and total protein ubiquitination is decreased during aging, and total protein ubiquitination is decreased during aging in lower animals such as C. elegans(Koyuncu et al, 2021), consistent with our findings", or "functional site of PQBP5/NOL10(Jin et al, 2023). To determine if similar sequestration of PQBP3/NOL3 to polyQ. To determine if similar sequestration of PQBP3/NOL3 to polyQ inclusion bodies occurred, we examined intracellular localization patterns of polyQ disease proteins and PQBP3/NOL3 in their coexpression (Figure 8a, b)".

>>> We are sorry for these errors that occurred during manuscript shuttling between us and professional English editors. We corrected these errors.

- using immunocytochemistry and immunoprecipitation (Figure 7a).
using immunocytochemistry and immunoprecipitation (Figure 7a).

>>> We corrected the sentence of this phrase.

- PSME3 appears to degrade proteins in a ubiquitin-independent manner (for review please see Mao et al Cell. Mol. Life Sci. 65, 3971-3980 (2008)). Can the authors discuss how this could fit with their data in Figure 7?

>>> We added experiments in Figure 9, and revealed that SUMOylated rather than ubiquitinated Lamin B1 is the target of PSME3-activated proteasomal degradation. The two type of protein modification occurred simultaneously on Lamin B1, which made their distinction difficult, whereas inhibitor experiments delineate the role of SUMOylation in PSME3-activated proteasomal degradation. We also discussed this issue in Discussion.

Minor points:

- The Introduction is very comprehensive, but it contains some information that it is not so relevant to understand the present study. The authors might think about shorten it to make it more straightforward to the topic of the study.

>>> Thank you very much for this kind evaluation and advice for Introduction. We thought carefully how we can shorten the introduction. As reviewer #2 mentioned, senescence is a topic of this paper and we need to keep the section for it. Also reviewer #3 asked us to discuss various types (or causes) of senescence. So we need

the introduction of senescence. Also we need to the background of this study, i.e. how we found PQBP3 and what the PQBP3 is.

An only part that we might be able to delete is the section describing PQBP3-cancer relationship. However, this is also related to cell proliferation that is reversely linked to senescence. We thought about moving the section to discussion. However, it disturbed the flow of discussion.

After thinking various modification, we could only partially shorten the introduction for cancer.

- Along these lines, I don't think the authors need to spend almost an entire paragraph to explain the differences between IUPred2A and RONN (a software that it's no longer available and the authors couldn't use). I would just focus on the analysis and conclusions from IUPred2A and remove all the discussion about RONN.

>>> We followed the advice and deleted discussion about PONN from the text. But since reviewer #3 strongly requested us to delete AFM and related data of IDPs, we decided to delete all these parts including RONN and also IUPred2A.

- After talking about the analysis using IUPred2A, they directly start the following paragraph with: "Glutathione S-transferase (GST) protein was cleaved from the GST-PQBP3 fusion protein (GST-PQBP3), and the cleaved PQBP3/NOL7 protein was subjected to high-speed atomic force microscopy (HS-AFM) for direct observation of dynamic protein structure changes". A better connection between paragraphs or context would help the readability of the paper. (for instance, starting the sentence with Glutathione S-transferase (GST) protein might be not so straightforward)

>>> This part was deleted due to the strong objection of reviewer #3 for AFM data.

- Recent studies have showed a role of PSME-activated proteasomes in the degradation of misfolded and disease-related proteins which are prone to misfold and aggregate including polyQ-expanded proteins (Frayssinhes et al, Cell. Mol. Life Sci. 2022, Lee et al, Nature Aging 2023). Here the authors discovered a link between PSME3, lamin B1 degradation and polyQ diseases. Given the role of PSME3 in degrading polyQ-expanded proteins, it would be interesting if the authors discuss these findings in the context of their results and the potential links with senescence.

>>>> We added a paragraph in Discussion, which is related to the advice of the reviewer.

Referee #3:

Jin et al. present an interesting study investigating the relationship between the nucleolus and senescence, specifically pertaining the role of PQBP3/NOL7 in protecting cells against senescence. The authors propose that loss of NOL7 function affects LaminB1, leading to cytoplasmic DNA accumulation which drives the senescence phenotype. They further suggest NOL7 is sequestered in ATXN1 aggregates in models of spinocerebellar ataxia, and that therefore Purkinje cells may suffer a similar fate. Overall, the experimental work is rigorous-though should be more quantitative in nature-and the manuscript is well-written. I do have some issues with the work in its current form, which I address in detail below. If the authors are able to address these comments, I would be supportive for publication of the work in EMBO journal.

>>> We really thank kind evaluation of our manuscript by the reviewer, and appreciate very much for helpful comments.

Major comments:

1. The study falls a bit short on the discussion and assessment of senescence. As the authors are aware, there are many different paradigms and discrepancies in the field. The choice of system should be discussed in more detail and characterized more. Neuronal senescence occurs not only due to normal aging, but also due to damage and other stress stimuli. Therefore, it is important for the authors to define and discuss their definition of senescence. For example, the authors assess contact inhibition and assume cellular quiescence/senescence, but there are better ways to define a

quiescent or senescent cell on a single-cell level. For example, the nuclear to cytoplasmic ratio of DHB is often used as an indication of G0-as DHB is nuclear during quiescence. Since the study aims to understand neuronal senescence in disease it is important to study if the same molecular interactions in damage-induced senescence and not only replicative senescence.

>>> We appreciate the thoughtful comments from the reviewer. It is true that there are many paradigms and definitions about senescence, and that cell damage such as DNA damage is a trigger of neuronal senescence. In the former part of this paper, we started the paradigm of quiescence/senescence. In the latter part of our paper, we showed that the similar change of PQBP3 to that in quiescence/senescence occurred in SCA1 neurodegeneration models, whose DNA damage our group has been proving to be enhanced (Qi et al, Nature Cell Biol 2007; Barclay et al, Hum Mol Genet 2012; Ito et al, EMBO Mol Med 2015; Taniguchi et al, Hum Mol Genet 2016). We have shown the similar change of PQBP3 in not only replication senescence but also in DNA damage-induced senescence. The final experiments in Figure 10 support the similar interaction could occur also in damage-induced senescence. We added these information in the result section.

>>> To support this idea, the kind advice regarding nuclear to cytoplasmic ratio of DHB was very useful, and we appreciate it very much. We performed analysis of N/C ratio with DHB-mVenus to show proliferation quiescence, hydrogen peroxide-induced senescence and Atxn1-induced phenotype were basically the same from the aspect of DHB-based evaluation of cell cycle (new Fig 10c).

Additionally, the idea that mTOR activity can define senescence versus quiescence is not specific enough. mTOR is implicated in

many pathways and the fluctuation of its activity cannot define the state. Although high mTOR activity may be true for quiescent cells and mTOR may increase in senescence, other markers in tandem within the same cell would be required to confidently define senescence versus quiescence.

>>> We performed beta-Gal staining in HeLa cells and human iPSC-derived neurons, also in response to other reviewer, whose quantitative states helped us to distinguish senescence versus quiescence. In addition, we detected SAHFs in senescence state cells by H3K9me3 staining (new Figure 6).

>>> Meanwhile, as another reviewer requested, we used an mTOR activator and found mTOR activation induced the similar change of PQBP3 subcellular distribution (new Figure 4) although mTOR is implicated in many signaling pathways as the reviewer kindly pointed out.

2. Figure 1 is distracting and completely irrelevant for this study. Whether NOL7 is a disordered protein or not contributes nothing to the rest of the work. The authors simply show some disorder prediction plots. The AFM data also does not add, but raises A LOT of questions. We see one supposedly representative example of a tadpole structure. What this means for NOL7's role in senescence is unclear. Whether this structure is physiological is unclear. The authors have the choice: (1) they omit the entirety of Figure 1 from the work as it adds no value, is distracting and highly preliminary or (2) they perform an in-depth structural characterization of NOL7 and link its disorder to its actual function. The authors have a great story here, but this figure detracts from the entirety of the work. I would strongly advise the authors to cut this data and pursue such studies in follow-up work. IDPs have been studied for decades. Showing that

a protein is an IDP without studying the functional consequences of that is not something that is of interest to the field in or benefits this study . I also disagree with having the very preliminary AFM data piggyback on the rest of the work. There are too many questions here for it to be published in its current form.

>>> We considered carefully about the suggestion that we should omit AFM and relevant results, and decided to follow the advice. We deleted these results.

3. Figure 4 is hard to interpret because of the DIC overlay. In general, the authors should also perform actual quantifications of the phenotypes they observe, rather than simply relying on representative images.

This is true for several figures in the manuscript. For example, here the authors should quantify nucleolar and cytoplasmic NOL7. Also, beta galactosidase should be quantified and correlated with nucleolar NOL7.

>>> Following the added the quantitative data in Figure 4 (new Figure EV2). We also quantified b-Gal stains in related experiments.

4. The authors discuss SAHFs as being an indispensable phenotype for senescent cells, but they do not confirm that a knockdown of NOL7 induces this phenotype. In prior figures, they only measured senescent markers with replicative senescence, but here we are looking at the supposed senescence induced by NOL7 knock-down.

>>> We performed Hoechst/H3K9Me staining of PQBP3/NOL7-KD cells and confirmed SAHFs phenotype (new Figure 6).

5. Figure 7: The authors should validate that their hydrogen peroxide protocol induces senescence by evaluating established markers.

>>> Following the advice, we checked b-Gal and SAHFs in HeLa cells and iPSC-derived neurons treated with hydrogen peroxide. Their results were consistent with senescence as expected.

6. Figure 8: This figure is somewhat puzzling. In cells we do see NOL7 recruitment to ATXN1 assemblies, but this is not seen (or not clearly shown) in ATXN1 mice. The authors discuss in length that the sequestration of NOL7 in polyQ aggregates is a novel toxic mechanism in the conditions, yet currently they do not provide any evidence for this sequestration. Overall, the images here should also be quantified, rather than just showing a single representative example.

>>> We thank the critical comment. We performed co-staining of Atxn1 and PQBP3 with ATXN1-KI mice, and confirmed their co-localization/sequestration. We also quantified the signal intensities of PQBP3 merged with or non-merged with Atxn1 and the signal intensities of PQBP3 located at nucleolus. These quantifications revealed changes of the PQBP3 signals in the presence of mutant Atxn1, and indicated that PQBP3 is sequestered to mutant Atxn1 foci.

7. Discussion: "Elucidating the full mechanism of the pathology like "digital twin" at the molecule or cell level could be accomplished using mathematical simulations based on integration of comprehensive data from multiomics and the LLPS-based interactome, which could be advanced from a prototype based on the proteome(Jin et al, 2021a)." This sentence is non-sensical and pretty much just a bunch of buzzwords thrown together, boiling down on

the vague promise that one day computer simulations will recreate reliable and useful models of cells that allow us to supposedly solve disease. I have literally no clue what the authors mean with this statement or how it relates to any of the work. What is an "LLPS-based interactome" even? As someone who has studied this phenomenon for nearly a decade I have no idea what the authors are referring to. Please remove, it is off-putting and has zero relevance to the data and the rest of the manuscript.

>>> We agree to remove the sentence, which we just wanted to describe, based on our own knowledge, that would be puzzling for others.

Dear Dr Okazawa,

Thank you for submitting your revised manuscript (EMBOJ-2023-114823R) to The EMBO Journal. Please accept my sincere apologies for the unusual delay in the process. Your amended study was sent back to the three referees for their scientific re-evaluation, and we have received detailed comments from two of them, which I enclose below. Please note that while reviewer #1 was not able at this time to reassess your work, we have editorially assessed your response to the critique and found the issues raised to be addressed satisfactorily. As you will see from the comments enclosed below, the other experts states that the work has been substantially improved by the revisions and they are now in favour of publication.

Thus, we are pleased to inform you that your manuscript has been accepted in principle for publication in The EMBO Journal.

We now need you to take care of a number of issues related to formatting and data presentation as detailed below, which should be addressed at re-submission.

Please contact me at any time if you have additional questions related to below points.

As you might have seen on our web page, every paper at the EMBO Journal now includes a 'Synopsis', displayed on the html and freely accessible to all readers. The synopsis includes a 'model' figure as well as 2-5 one-short-sentence bullet points that summarize the article. I would appreciate if you could provide this figure and the bullet points.

Thank you for giving us the chance to consider your manuscript for The EMBO Journal. I look forward to your final revision.

Again, please contact me at any time if you need any help or have further questions.

Best regards,

Daniel Klimmeck

>> Please add up to five keywords to the manuscript text.

>> Author Contributions: Please remove the author contributions information from the manuscript text. Note that CRediT has replaced the traditional author contributions section as of now because it offers a systematic machine-readable author contributions format that allows for more effective research assessment. and use the free text boxes beneath each contributing author's name to add specific details on the author's contribution.

More information is available in our guide to authors.
<https://www.embopress.org/page/journal/14602075/authorguide>

>> Adjust the title of the 'Competing Interests' section to 'Disclosure and Competing Interests Statement' and move it after Acknowledgements.

>> Funding: please mention the 'Grant-in-Aid for Scientific Research A (16H02655) from the Japanese Society for the Promotion of Science (JSPS)' in our online system. Funding should be included in the section "Acknowledgments".

>> Please provide a comprehensive set of source data for the study as to the separate request e-mail by my colleague Hannah Sonntag.

>> Consider additional changes and comments from our production team as indicated below:

>> Callouts: add a callout for Fig.5E in the running text.

>> Indicate redisplay of cellular data from Fig7B and within Fig. EV3 in the Figure legend of Fig. EV3.

- Figure Legends (main + EV):

1. Please note that the legend for Figure 4e is missing in the manuscript. This needs to be rectified.
2. Please note that in the legend for Figure EV 2a, the red arrows are labelled as yellow arrows. This needs to be rectified.
3. Please define the annotated p values ##### as well as provide the exact p-values for the same in the legend of figure 5a, e; 9e; as appropriate.
4. Please note that the exact p values are not provided in the legends of figures 3b-e; 5d; 6b, d; 8b, d-e; 9d; EV 2g.
5. Please indicate the statistical test used for data analysis in the legends of figures 5a, e; 9e.
6. Please note that in figures 6b, d; there is a mismatch between the annotated p values in the figure legend and the annotated p values in the figure file that should be corrected.
7. Please note that the box plots need to be defined in terms of minima, maxima, centre, bounds of box and whiskers, and percentile in the legends of figures 2c; 3b-e; 4c; 5a, d-e; 6b, d; 8b, d-e; 9d-e; 10c, f; EV 1a, c; EV 2g.
8. Please note that information related to n is missing in the legends of figures 2c; 4c; 5a; 8d; 9e; EV 1c.
9. Please note that scale bar and its definition are missing for figures 2a; EV 1b; EV 2b-f, h-j; EV 3.
10. Please note that the black arrow is not defined in the legend of figure 9a. This needs to be rectified.
11. Please note that the white arrows are not defined in the legends of figures 5c, 7a, EV 2e-f, h-j. This needs to be rectified.
12. Please note that the white and yellow arrows are not defined in the legends of figures 8b. This needs to be rectified.
13. Please note that the black and red arrows are not defined in the legends of figures 8c, 9e. This needs to be rectified.
14. Please note that the blue/white/gray arrows are not defined in the legends of figures 10d-f. This needs to be rectified.

Referee #2:

The authors have addressed all my comments and I support publication of the article.

Referee #3:

The authors have addressed all comments raised by this reviewer.

The authors addressed the minor editorial issues.

Dear Dr Okazawa,

Thank you for submitting the revised version of your manuscript. I have now evaluated your amended manuscript and concluded that the remaining minor concerns have been sufficiently addressed.

I am thus pleased to inform you that your manuscript has been accepted for publication in the EMBO Journal.

On a different note, I would like to alert you that EMBO Press offers a format for a video-synopsis of work published with us, which essentially is a short, author-generated film explaining the core findings in hand drawings, and, as we believe, can be very useful to increase visibility of the work. Please see the following link for representative examples and their integration into the article web page:

<https://www.embopress.org/doi/full/10.15252/emj.2019103932>

If you have any questions, please do not hesitate to contact the Editorial Office.

Best regards,

Daniel Klimmeck

Daniel Klimmeck, PhD
Senior Editor
The EMBO Journal
EMBO
Postfach 1022-40
Meyerhofstrasse 1
D-69117 Heidelberg
contact@embojournal.org